



# Lead fractions from SAR-derived sea ice divergence during MOSAiC

Luisa von Albedyll[1], Stefan Hendricks[1], Nils Hutter[2,1], Dmitrii Murashkin[3,4], Lars Kaleschke[1], Sascha Willmes[5], Linda Thielke[4], Xiangshan Tian-Kunze[1], Gunnar Spreen[4], and Christian Haas[1,4]

[1]Alfred-Wegener-Institut, Helmholtz-Zentrum für Polar- und Meeresforschung, Bremerhaven, Germany
[2]University of Washington, Cooperative Institute for Climate, Ocean, and Ecosystem Studies (CICOES), Seattle, WA, United States
[3]German Aerospace Center (DLR), Remote Sensing Technology Institute, Bremen, Germany
[4]University of Bremen, Institute of Environmental Physics, Bremen, Germany
[5]Trier University, Dpt. Environmental Meteorology, Trier, Germany

**Correspondence:** Luisa von Albedyll (luisa.von.albedyll@awi.de)

**Abstract.**

Leads and fractures in sea ice play a crucial role in the heat and gas exchange between the ocean and atmosphere, impacting atmospheric, ecological, and oceanic processes. Our aim was to estimate lead fractions from high-resolution divergence obtained from satellite synthetic-aperture radar (SAR) data and to evaluate it against existing lead products. We derived two new lead-

fraction products from divergence with a spatial resolution of 700 m calculated from daily Sentinel-1 images. For the first lead product, we advected and accumulated the lead fractions of individual time steps. With those accumulated divergence-derived lead fractions, we described comprehensively the presence of up to 10-day-old leads and analyzed their deformation history. For the second lead product, we used only divergence pixels that were identified as part of linear kinematic features (LKFs). Both new lead products accurately captured the formation of new leads with widths of a few hundred meters. We presented a

Lagrangian time series of the divergence-based lead fractions along the drift of the MOSAiC expedition in the central Arctic Ocean during winter 2019/2020. Lead activity was high in fall and spring, consistent with wind forcing and ice pack consolidation. At larger scales of 50-150 km around the MOSAiC expedition, lead activity on all scales was similar, but differences emerged at smaller scales (10 km). We compared our lead products with 6 others from satellite and airborne sources, including classified SAR, thermal infrared, microwave radiometer, and altimeter data. We found that the mean lead fractions varied by

1 magnitude across different lead products due to different physical lead and sea ice properties observed by the sensors and methodological factors such as spatial resolution. Thus, the choice of lead product should align with the specific application.

## 1 Introduction

Divergent motion in sea ice leaves open water in the sea ice cover, which we refer to as fractures or leads. These openings play

a crucial role in the polar climate system altering atmospheric, ecological, and oceanic processes. In winter, the exchange of gases and heat between the ocean and atmosphere is strongly enhanced at the openings in the ice in the otherwise well-separated





components of the polar climate system (Maykut, 1978, 1982; Perovich, 2011). Turbulent heat is transferred from the ocean to the atmosphere, followed by fast new ice formation, and brine rejection to the ocean during winter. The enhanced exchange has important implications. First, new ice formation in leads contributes about 30% to the Arctic sea ice mass balance and

the thin ice influences the sea ice dynamics (von Albedyll et al., 2022; Kwok, 2006). Second, they enable ocean-atmosphere gas exchange, for instance, water vapor, iodine, and methane relevant in Arctic cloud formation (e.g., Leck et al., 2002; Kort et al., 2012; Dall´Osto et al., 2017). Third, they may act as sources of atmospheric sea salt from frost flowers growing on ice-covered leads (e.g., Perovich and Richter-Menge, 1994; Kaleschke et al., 2004; Hara et al., 2017). Forth, in summer, their low albedo increases solar transmission to the ocean. Fifth, they act as important hunting grounds for marine mammals, and

sixths, they are important shipping routes (e.g., Massom, 1988; Stirling, 1997). In addition, leads are easily detectable signs of sea ice deformation, and studying their occurrence, spacing, orientation, intersection, and scale invariance is of great relevance for sea ice mechanics (e.g., Weiss and Marsan, 2004; Hutter et al., 2019; Hutter and Losch, 2020; Ringeisen et al., accepted). Lastly, leads are important for remote sensing of sea ice thickness by satellite altimetry as the method relies on lead detections to measure the instantaneous sea surface height for ice freeboard retrieval (e.g., Laxon et al., 2003).

The distribution of leads follows the overall dynamic regimes of the Arctic Ocean. While the lead frequency, i.e., how often a lead is found at a certain position within a certain time period, is a few percent for the central Arctic Ocean, it can rise to over 40% in the Barents and Kara Seas, and the marginal ice zone (Willmes and Heinemann, 2016; Reiser et al., 2020). Although leads cover only 1-3% (Wadhams, 2000; Reiser et al., 2020), their impact on the winter heat budget is significant (Maykut, 1978; Marcq and Weiss, 2012): An increase of 1% in lead fraction can increase the near-surface temperatures in the Arctic by

3.5 K under clear sky conditions during Polar Night (Lüpkes et al., 2008).

Due to their high relevance for the polar climate, accurate knowledge of the location and areal fraction of leads ("lead fraction") is crucial for understanding and modeling processes on the air-ocean interface, but also the global weather and climate (Serreze et al., 2009). This is particularly interesting considering that Arctic sea ice drift and deformation rates are increasing (e.g. Rampal et al., 2009; Spreen et al., 2011), with yet unclear impact on the changes in lead fractions. Consequently, the importance

of leads for the sea ice mass balance has gained increased attention in recent sea ice modeling studies (e.g. Wilchinsky et al., 2015; Zhang et al., 2021; Ólason et al., 2021; Boutin et al., 2023), as well as using lead statistics to evaluate different rheological frameworks for sea-ice dynamics (Wang et al., 2016; Hutter and Losch, 2020; Hutter et al., 2022). In contrast to the transient nature of leads, they can have a long deformation history as the ice breaks preferentially where it is thinner than the surrounding ice (e.g., Wilchinsky and Feltham, 2011). In other words, leads can undergo several cycles of opening, closing, and ridging,

interrupted by dormant phases ranging from days to months. However, there is a general lack of high-resolution reference datasets available for evaluating the magnitude, temporal and spatial variability, and deformation history of leads.

Satellites have monitored the spatial distribution and temporal evolution of Arctic lead fraction since the 1990s (Key et al., 1993; Lindsay and Rothrock, 1995; Miles and Barry, 1998). Lead detection based on thermal infrared (TIR, Willmes and Heinemann, 2016; Hoffman et al., 2022; Wang et al., 2022; Qiu et al., 2023) and visible images (e.g., Lewis and Hutchings, 2019; Muchow

et al., 2021) was complemented by classification of Synthetic Aperture Radar (SAR) backscatter (e.g., Murashkin et al., 2018),





radar altimeters (e.g., CryoSat-2 Hendricks et al., 2021a), laser altimeters (e.g, ICESat-2 Duncan and Farrell, 2022), and passive microwave (PMW) data (e.g., Röhrs et al., 2012). The transient nature of leads and their narrow appearance sets limits to the detection of leads from satellites. Most retrieval methods suffer either from a low spatial resolution (e.g., PMW), low spatial coverage (e.g., altimeters), low temporal coverage due to clouds (e.g., TIR and visible), or ambiguous classification due to

acquisition geometry (e.g., SAR). In addition, the definition of a lead is ambiguous as it can be covered by open water or thin ice of up to 30 cm thickness (new and young ice according to World Meteorological Organization (2014)). However, the different lead identification methods do not have a clear boundary when the ice gets too thick to be classified as lead. This results in inconsistent estimates of the lead fraction between the retrieval methods that can vary by magnitudes. So far, there are only a few comparison studies between lead products, which concluded that the compared products show often similar

spatio-temporal patterns but vary substantially in magnitude (Kwok, 2002; Lee et al., 2018).

In lead-fraction retrievals, little focus has been placed on deriving lead fractions from their driving mechanism, i.e., the divergent motion of the ice floes (Kwok, 2002). The sea ice divergence of the sea ice velocity is directly linked to lead fraction and can be calculated from high-resolution sea ice drift. Kwok (2002) demonstrated the technique using RADARSAT Geophysical Processor System (RGPS) data, in the Pacific Sector of the Arctic with a 3-day temporal resolution. So far, the low temporal

resolution of SAR images hindered a more area-wide application, but this has changed in recent years with the start of the Sentinel-1 constellation and many other SAR missions. This growing availability of SAR images in the polar region motivates us to explore the potential of divergence derived from SAR data for estimating lead fraction. More precisely, here we present a novel lead fraction dataset based on data from the Sentinel-1 constellation.

The interdisciplinary Multidisciplinary drifting Observatory for the Study of Arctic Climate (MOSAiC) expedition took place

between October 2019 and September 2020 in the Transpolar Drift on board the R/V *Polarstern* (Nicolaus et al., 2022; Rabe et al., 2022; Shupe et al., 2022). We have selected the MOSAiC expedition as our study case for several reasons. First, the drift covered a wide range of different dynamic regimes (Krumpen et al., 2021). Second, there are important complementary atmospheric and ocean datasets and regional, high-resolution, airborne observations of leads available. Third, leads were a cross-cutting theme for all MOSAiC disciplines, i.e., ice, ocean, atmosphere, ecology, and bio-geo-chemistry, and a sound

estimate of regional lead fractions is crucial for their research (Nicolaus et al., 2022; Rabe et al., 2022; Shupe et al., 2022).

This study has two objectives. First, we aim to present and evaluate two novel lead products that are based on SAR-derived sea-ice divergence. To do so, we compare the novel lead products with six, already existing lead products. Our second aim is to present and analyze a time series of divergence-based lead fractions along the MOSAiC drift track. As most lead products are only available in winter time, we restrict our analysis to October 2019 to May 2020.

The structure of the study is as follows: In Section 2, we describe the physical properties of leads used to detect them and introduce the novel lead products along with the existing ones participating in the comparison. Section 3 presents the properties of the lead products based on divergence including basic statistics of the observed leads and a time series on different spatial scales. Section 4 analyzes the temporal and spatial differences between the novel and existing lead products. The concluding sections 5-6 discuss and summarize the results and outline potential improvements.



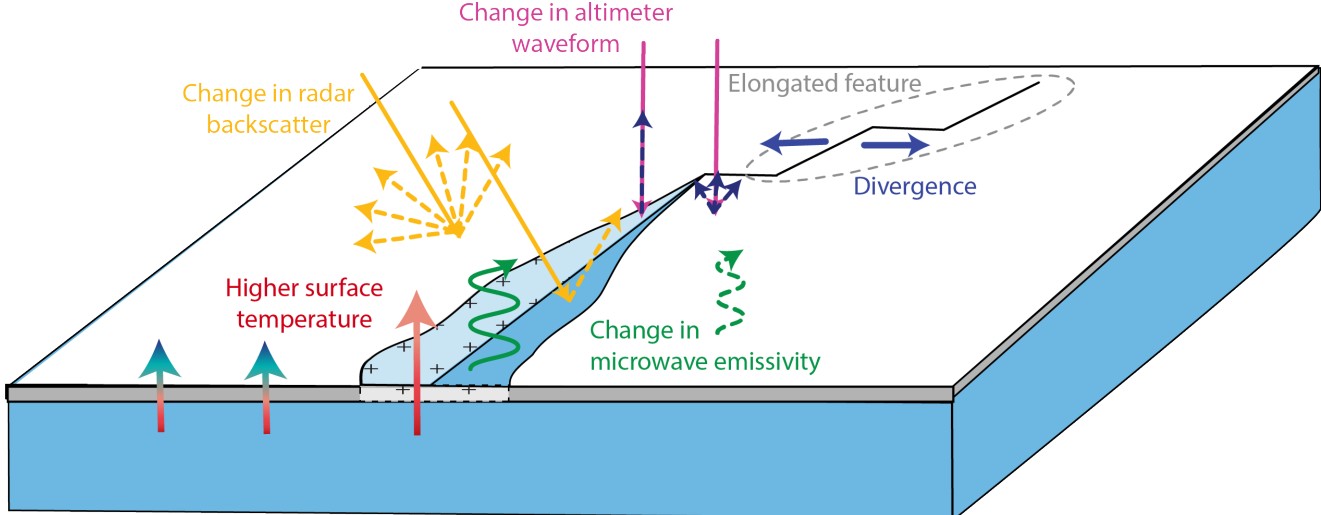

**Figure 1. Schematics of different physical properties of a lead detected by remote sensing instruments.**

## 2   Lead fractions from different retrieval methods

What is a lead? Each lead-fraction retrieval method gives a different answer to this question. Depending on the application and the research question behind it, the underlying lead definition of a lead product may have advantages or restrictions. Therefore, we start this methods and data section with a description of the physical properties of a lead (Section 2.1) and explain next, how the different retrieval methods make use of them (Sections 2.2–2.3).

### 2.1   Physical properties of a lead detected by remote sensing

The World Meteorological Organisation (WMO) defines a lead as a "fracture or passage-way through sea ice which is navigable by surface vessels" whereby fractures are defined as "break[s] or rupture[s] through very close ice, compact ice, consolidated ice [...] resulting from deformation processes. Fractures may contain brash ice and/or be covered with nilas and/or young ice" (World Meteorological Organization, 2014). Following this definition, a lead can be covered by up to 30 cm (young ice) thick ice (World Meteorological Organization, 2014). In this study, and almost everywhere else in literature, fractures and leads are often summarized under the term leads, and their minimum width and maximum ice thickness depend on the sensitivity of the retrieval method. However, this sensitivity, especially with respect to the maximum lead ice thickness is often not precisely known.

Figure 1 provides an overview of the physical properties of leads that are outlined as follows:

(1) **Local change in ice velocity**

Leads result from deformation processes and can be detected by a strong local gradient in the ice velocity. External drivers, mainly winds and ocean currents induce stress on sea ice. Sea ice breaks when those stresses reach the ice





strength, resulting in deformation (for an overview see Weiss, 2013). Breaking, followed by divergent ice motion forms leads and the divergence magnitude is directly proportional to the lead width.

(2) **Elongated features**

Leads have typically an elongated shape with a long extent in one direction (length) and a very small extent in the other direction (width). In the absence of coastlines, they can appear in systems of parallel faults in the order of several kilometers to thousand kilometers (Goldstein et al., 2000).

(3) **Abrupt change in surface properties**

Leads have very different surface properties than the surrounding ice with respect to the roughness, salinity, surface temperature, and surface height probability distribution function. The scattering and absorption of radar (microwave) waves depend strongly on those parameters and therefore leads typically provide a strong contrast to the surrounding sea ice surface. For example, on SAR images, the reduced roughness in leads with calm open water or a smooth, new ice cover has a low radar backscatter signature. Waves on open water or frost flowers on thin lead ice can quickly turn a lead

into a very rough, and thus stronger scattering surface. The abrupt change in the surface properties also modifies the shape of the altimeter radar waveforms in a characteristic way which is used to detect leads by, e.g., CryoSat-2 (e.g., Wernecke and Kaleschke, 2015; Paul et al., 2018). In optical remote sensing, leads appear darker due to specular reflection on the smoother surface and light transmission through the thin ice into the open water. However, visible images are not included in this study because we focus on the wintertime and high latitudes, where no sunlight is available.

(4) **Higher surface temperature**

In winter, the ocean temperature in leads is at the freezing point close to –1.8 °C, while the surrounding ice and snow surface approach the much colder air temperatures, resulting in temperatures differences of –10 to –40°C. This tempera-ture contrast is well seen in the TIR part of the EM spectrum. When new ice starts to form in the open water, the surface temperature anomaly in leads gradually decreases.

(5) **Special microwave emissivity**

Thin ice has a special emissivity in the microwave spectrum. Comparing the emissivity at different frequencies and polarizations allows us to detect open water and thin ice. While the polarization difference is highest for open water, leads covered with thin ice exhibit a particularly high emissivity at 89 GHz with vertical polarization (Eppler et al., 1992). The lead signature is further enhanced using a high pass filter to derive the thin ice fraction with a tie-point

method. (Röhrs et al., 2012; Ivanova et al., 2016).

(6) **Local thickness and surface elevation minimum**

Leads are characterized by open water or thin ice and thus exhibit a local minimum in the ice thickness and surface elevation. When the lead is not closed by convergent dynamics, this difference to the surrounding ice can persist. How-ever, there is no agreed-upon maximum thickness for considering the thin ice still as a lead. Thresholds in different

studies range between open water and 1 meter (Wadhams, 2000). In this study, we only use the criteria of a local surface





elevation minimum to define leads in the airborne ice thickness observations, a dataset that is used for complementary comparisons and is available here: von Albedyll et al. (2021b, von Albedyll et al. (2022)). However, there are sensors that make use predominantly of the surface elevation minimum to detect leads, e.g. the laser altimeter ICESat-2.

The following sections introduce the specific characteristics and retrieval methods of two new and six existing lead products
that exploit one or several of the above-mentioned properties of leads to detect them.

## 2.2   Novel lead product based on SAR-derived sea ice divergence

In this section, we present two new lead products. These are based on divergence in the sea ice motion that is derived from SAR data with a spatial resolution of 50 m. The divergence indicates the exact location of the lead, is independent of clouds and, as noted by Kwok (2002), is independent of sensor calibrations and physical understanding of the radar backscatter signal
of the ice.

While Kwok (2006) has previously used this general concept, we present two products that have a much higher temporal and spatial resolution, are regularly gridded, are advected and accumulated over several days, and are evaluated extensively with other lead products and a complimentary, airborne ice thickness dataset. The first lead product is based solely on the divergence, while the second product derives at first linear kinematic features (LKFs) from the total deformation before indicating the
presence of leads.

### 2.2.1   Accumulated divergence-derived lead fractions ($LF_{accu. div}$)

Our first novel lead product is directly derived from the divergence fields that are calculated following von Albedyll et al. (2021a). The divergence-derived lead fractions ($LF_{div}$) detect the strong local change in ice velocity. Due to the direct relationship, divergence results directly in lead formation.

Calculating lead fractions from divergence consists of three steps. First, we derive sea ice drift and deformation fields. We use sequential SAR scenes in HH polarisation Extra Wide swath mode obtained by the Sentinel-1 mission (ESA) with a spatial resolution of 50 m (Torres et al., 2012). The scenes were acquired along the drift track of the MOSAiC expedition and cover roughly an area of 150 km around R/V *Polarstern*. We aim for a nominal time step of one day between the scenes but accept everything between 0.9-3 days. The lower limit of 0.9 days is chosen to guarantee a displacement larger than the uncertainty
of the tracking. Images are available for the entire study period from October 2019 to May 2020, except for the time between 14 January and 15 March 2020, when the ship was north of the latitudinal limit of Sentinel-1. We use a tracking algorithm based on Thomas et al. (2008, 2011) and substantially extended by Hollands and Dierking (2011) to derive sea ice drift. Next, we calculate the spatial derivatives from the regularly spaced drift field following von Albedyll et al. (2021a). Divergence and convergence are then derived from the spatial derivatives of the velocity field $(u, v)$:

$$\text{div} = \frac{\partial u}{\partial x} + \frac{\partial v}{\partial y} \tag{1}$$





Divergence ($\mathrm{div} > 0$) and convergence ($\mathrm{div} < 0$) are defined as positive and negative $\mathrm{div}$, respectively. We filter the deformation data with a directional filter that detects the direction with the smallest variation at each pixel and smooths along, but not across this orientation, with a 1-d kernel. The gradients in deformation along the 1d-kernel are calculated in the form of the standard deviation for a neighborhood of 7 pixels. This way, noise is reduced while preserving the strong gradients in the velocity field

that are indicative of deformation. The MOSAiC divergence fields with a final resolution of $700\,\mathrm{m}$ were previously used in Krumpen et al. (2021) with the difference that here we use a directional filter instead of the 3x3 median filter to reduce noise (see also von Albedyll et al., 2021a; Ringeisen et al., accepted).

Second, for the calculation of the lead fractions, we multiply all divergence grid cells [$div$ in s$^{-1}$] by the time step length [in s], resulting in a unit-free lead fraction. Since the divergence quantifies the relative area expansion per time step, this results

in the relative area expansion for each grid cell, and thus an estimate of the area that is covered by open water. We interpret the result as the average lead fractions per grid cell. Please note, only positive values (divergence) are lead fractions, as the negative values indicate closing and ridging. However, we still keep the convergence information as we require it in the next step.

Up to this point, the lead fraction algorithm only detects leads when they form or continue to open. To determine if a lead is

closed, opened further, or stayed open, we use the divergence/convergence fields of subsequent dates. In other words, as the divergence-derived lead fractions describe the change in lead fraction, accumulating them describes the dynamic evolution of a lead over several days. To account for the movement of sea ice, we need to advect the lead fractions from different dates to a common location using sea ice drift data prior to accumulating them. Fortunately, sea ice drift is inherently available as the divergence is calculated based on it.

We perform three steps to calculate accumulated divergence-derived lead fractions (LF$_{\mathrm{accu.\ div}}$). Figure 2 displays an example from March 27, 2020.

    (1)  We standardize the drift and divergence fields to a common grid based on the polar stereographic projection for the northern hemisphere (EPSG:3413) with a spatial resolution of $700\,\mathrm{m}$.

    (2)  b1 each lead-fraction grid cell to its new location based on the respective displacement and re-grid the field back to the

original grid. We repeat this procedure for 10 time instances. For example, we advect lead fractions based on two SAR images from March 14/15, to their respective location on March 15, March 16, March 17, March 18, March 19, March 20, (gap in SAR images March 21 and 22), March 23, March 24, March 25, March 26, and March 27. Next, b1 the lead fractions originally based on SAR scenes from March 15/16 to its location on March 16, March 17, ..., and March 28. We save all advected lead fractions of a time instance in one NetCDF file. This means the NetCDF file of March 27 contains

datasets of advected lead fractions from originally March 14/15, March 15/16, March 16/17, March 17/18, March 18/19, March 19/20, March 20/23, March 23/24, March 24/25, March 25/26, and March 26/27.

    (3)  We accumulate the advected lead fractions in each grid cell considering opening, closing, and dormant phases. The accumulation can start at any of the 10 time instances. This gives the data user the flexibility to choose for themselves





up to which "age" (1-11 days) they would like to include the leads of different ages in their analysis. The lead fraction
datasets still contain information on opening (positive values) and closing (negative values). We calculate the cumulative
sum of the lead fraction time instances starting with the earliest one. For each iteration of the accumulation, we check
whether the cumulative sum becomes negative. This corresponds to the full closure of the lead. At this point, we reset
the cumulative sum to zero.

To extract the mean (accumulated) lead fraction of a certain region, e.g., 50 km around R/V *Polarstern*, we calculate the average
of all grid cells of the advected accumulated lead fractions that are located completely or partly in a circle with 10-150 km radius
around R/V *Polarstern*. All grid cells indicating (accumulated) ridging are set to 0 prior to the averaging.

### 2.2.2 LKF-derived lead fractions ($LF_{LKF}$)

As our second new dataset, we compute LKF-derived lead fractions ($LF_{LKF}$). They are based on the deformation dataset (see
previous section) to which we applied the LKF detection algorithm developed by Hutter et al. (2019, details are given below).
As input, we use total deformation calculated from divergence and shear ($\sqrt{\mathrm{div}^2 + \mathrm{shr}^2}$). To reduce noise, the LKF detection
algorithm makes use of the localized nature of deformation and selects only those total deformation features that have a strong
velocity contrast and an elongated shape (Hutter et al., 2019). The output are unique LKFs.

The LKF detection algorithm (1) generates a binary mask of pixels identified as LKFs by filtering pixels with total deformation
rates that strongly exceed the local average deformation rates, (2) morphologically thins the LKFs in the binary map to a
width of one pixel, and divides it into the small linear segments, and (3) reconnects segments into LKFs based on their
similarity in position, orientation, and deformation rates. In the second step, the algorithm uses the drift information between the
deformation fields to track LKFs over time. The morphological thinning routine was modified to align the LKF (morphological)
skeletons in the binary maps to the position of the highest deformation rates across the LKF. For each LKF, the algorithm
outputs the position, the original divergence value at that location, and an ID of the LKF that preceded the current one if
available (Hutter et al., 2019).

We are computing two different quantities. Firstly, for the LFLKF lead fractions, we identify all instances where positive
divergence values appear within the detected LKFs (leads). These divergence values are then multiplied by the length of the
time step for the grid cell lead fraction. To find the average lead fraction over a larger area, we calculate the mean of all
grid cells that are either fully or partially situated within a circle with a radius of 10-150 km centered on the position of R/V
*Polarstern*.

Secondly, For the $LF_{LKF}$ binary lead pixel number, a pixel is considered completely covered by a lead when a positive diver-
gence value is present in the detected LKFs. This is identical to assuming a lead fraction of 1. However, in most cases, this
method will result in overestimating the actual lead fraction. This counterbalances that the LKF detection algorithm simplifies
all deformation zones into 1-dimensional features, effectively removing divergence information.

We do not accumulate the $LF_{LKF}$ in time. They only indicate the instantaneous presence of new leads from the time of one
SAR image to the next one.





### 2.3 Other existing lead products used for comparison

#### 2.3.1 Classified-SAR lead fractions (LF$_{classified\_SAR}$)

To generate classified-SAR lead fractions (LF$_{classified\_SAR}$), we apply the supervised learning classification algorithm that was
presented in Murashkin et al. (2018) and Murashkin and Spreen (2019) to SAR images along the MOSAiC drift track.

Lead classification of SAR images relies on the abrupt change in the surface roughness and salinity in leads that are observed as
changes in the radar backscatter signal. Leads with calm open water or thin ice are specular scatter targets in contrast to sea ice
surfaces with more diffuse scatter. Leads with waves or frost flowers have higher backscatter, especially when using radar with
a wavelength in the range of a few centimeters like Sentinel-1 (5.5 cm wavelength). Thus, in such cases, leads are characterized
by strong backscatter. Depending on the backscatter statistics (lower or higher than the backscatter of the surrounding sea ice),
this algorithm detects open water and leads covered with thin ice.

As input, the supervised learning classification algorithm uses both the HH and the HV SAR channels of Sentinel-1 Extra Wide
swath scenes. The use of the cross-polarization band allows the separation of rough surface leads and ridges, both of which are
characterized by a strong backscatter in co-polarization backscatter. The algorithm is based on SAR image analysis with the
UNET convolutional neural network (Ronneberger et al., 2015). The algorithm produces binary maps with a lead classification
with the same spatial resolution of 40 m as the source Sentinel-1 scenes.

We apply the classification algorithm to the same Sentinel-1 scenes that we use for the divergence calculations (Section 2.2.1),
even if there are more scenes available at sub-daily resolution. This offers us the best conditions for a comparison between
the three SAR-based datasets. We estimate the mean lead fraction by calculating the relative fraction of all pixels classified as
leads within a circle with a radius of 50 km around R/V *Polarstern*.

#### 2.3.2 MODIS lead fractions (LF$_{MODIS}$)

For our comparison, we use lead fractions derived from Moderate-Resolution Imaging Spectroradiometer (MODIS) ice surface
temperatures, Collection 6 (Hall and Riggs, 2019). The algorithm to derive the MODIS lead fractions (LF$_{MODIS}$) is described
by Willmes and Heinemann (2015) and Reiser et al. (2020). A time series of lead fractions for MOSAiC based on this dataset
was previously presented in Krumpen et al. (2021).

Based on the method described in (Reiser et al., 2020) a lead is detected by its significant positive surface-temperature anomaly
as compared to the colder surrounding sea ice. Open water and a thin sea ice cover create a positive surface-temperature
anomaly. False detections can arise due to unidentified clouds, fog, and sea smoke. They are minimized using shape-, per-
sistence, and texture metrics of potential leads in addition to surface temperature. A fuzzy filter is then applied using these
stacked metrics to assign individual retrieval uncertainties to each identified lead pixel. This results in daily binary lead maps
with cloud gaps at a spatial resolution of 1 km.

Each grid cell in the resulting daily lead maps can be classified as cloud, sea ice, lead, and artifact, with the latter comprising
detected leads with an uncertainty exceeding 30%. Due to the gridding, i.e. the mapping of a lead into a regular grid with a





grid size of 1 km, and the binary classification scheme that only allows a grid cell to be fully covered by a lead or not at all, the
actual lead fractions may be clearly overestimated.

We calculate lead fractions in the 50 km circle as the ratio of lead over all valid, i.e., leads plus sea ice, pixels. We exclude the
mean lead fraction from our analysis when the percentage of valid data in the 50 km circle falls below 50%.

### 2.3.3 Helicopter-borne TIR lead fractions (LF$_{\text{Heli\_TIR}}$)

We present lead fractions from nine regional helicopter survey flights that were conducted with a thermal infrared camera
(Thielke et al., 2022; Thielke et al., 2022; Thielke et al., in review). Deriving the helicopter-borne TIR lead fractions (LF$_{\text{Heli\_TIR}}$)
relies on the same principle of higher surface temperatures as the LF$_{\text{MODIS}}$ described in the previous section. However, the
LF$_{\text{Heli\_TIR}}$ have a much higher resolution of up to one meter and suffer less from the interference with atmospheric conditions
due to a low flight altitude of around 300 m (Thielke et al., 2022). Flights were performed only during clear and calm weather
conditions.

The nine regional helicopter survey flights used in this study were conducted between October 2019 and May 2020 at positions
along the drift track of MOSAiC (Thielke et al., 2022, in review). We use the broadband measurements of the mounted
thermal infrared camera from 7.5 to 14 $\mu$m which is in the similar frequency range as MODIS. To classify leads from the
measured surface temperature, we apply an iterative threshold selection method (Ridler and Calvard, 1978) on the gridded
surface temperature maps. The resulting products are binary maps for lead occurrence covering the flight tracks in up to
30 km distance to R/V *Polarstern*. Due to changing conditions, e.g. air temperatures, throughout the season, we use a dynamic
threshold, i.e., a different threshold for each flight. To calculate the final lead fraction in the 50 km circle, we divide the number
of pixels classified as lead by the number of all-valid pixels along the flight track.

### 2.3.4 Passive microwave lead fractions (LF$_{\text{PMW}}$)

In this study, we use PMW lead fractions (LF$_{\text{PMW}}$) derived from data collected by the Advanced Microwave Scanning Ra-
diometer 2 (AMSR-2) to which we apply an updated version of the algorithm previously introduced by Röhrs et al. (2012) for
the preceding instrument AMSR-E.

Lead fractions derived from satellite PMW imagery make use of the strong surface-emissivity contrast to distinguish between
leads and thick ice. To distinguish the elongated shape of leads from large areas of polynyas which are also covered by thin ice,
a high-pass filter is applied to detect the lead edges. In contrast to the TIR imagery, lead detection from PMW works largely
unaffected by clouds but is affected by melting conditions. From all products, the LF$_{\text{PMW}}$ resolve the presence of lead ice, but
not necessarily only thin ice, the longest. From the relatively long persistence of the lead features observable with the AMSR2
sensor, up to several weeks, we can conclude that the method detects not only typical lead ice types (new ice, nilas, young ice,
gray and gray-white ice) but also thicker ice, i.e. first-year ice (> 30 cm) formed in refrozen leads. Therefore, lead fractions can
be clearly higher than from other products. The greatest advantage of satellite passive microwave data is their high temporal
and spatial coverage and the length of the data record that spans from the 1980s to nowadays.



AMSR-2 is the follow-on instrument of AMSR-E with similar frequencies and spatial resolution. Lead fractions were previously derived from AMSR-E data by Röhrs et al. (2012) using vertically polarized brightness temperature ratio between the 89 GHz and 19 GHz channels that is distinctive for thin ice. The AMSR-E lead detection algorithm provides the estimation of thin ice fraction within an AMSR-E grid resolution, thus a mixture of leads with different sizes. The lead-fraction dataset was expanded to the AMSE-2 period by applying the same algorithm. However, the parameters in the algorithm had to be adjusted to the new instrument. To homogenize lead fractions from the two instruments, a cross-comparison with MODIS-derived lead fractions was carried out in a pre-study (Kassens et al., 2020). In the pre-study, the new tie points of the brightness temperature ratios which correspond to 0% and 100% thin ice concentrations/lead fractions were estimated in a 500 km x 500 km large area in the Beaufort Sea where frequent lead openings were observed. AMSR-2 swath brightness temperatures were gridded into NSIDC polar-stereographic projection with 3.125 km x 3.125 km grids, and lead fractions were calculated in each grid cell over the entire Arctic on a daily basis. For this study, we calculate mean lead fractions as the average of all grid cells in the 50 km circle.

### 2.3.5 CryoSat-2 lead fractions ($LF_{CS2}$)

For the CryoSat-2 lead fractions ($LF_{CS2}$), we use the "Level-3 gridded sea-ice thickness and auxiliary parameters" product (version 2.4) from the Alfred Wegener Institute Helmholtz Center for Polar and Marine Research (Hendricks et al., 2021a) based on calibrated CryoSat-2 sensor data (European Space Agency, 2019).

Satellite radar altimeters, like CryoSat-2, take advantage of the contrasting radar backscatter characteristics of leads whose flat surfaces result in a narrow radar waveform with a large amplitude compared to the wider and weaker waveforms of rougher sea ice surfaces. This dependency of radar waveform properties to surface type, e.g., the pulse-peakiness, is widely used to discriminate between sea surface and sea ice elevations and to estimate both sea surface height and sea ice freeboard (Quartly et al., 2019). In the context of sea ice altimetry, a lead can be covered by open water, but certainly also young ice such as nilas and grey ice. The strict nadir pointing of altimeters results in significantly less ambiguity of the radar backscatter signature over leads compared to other radar methods with oblique incidence angles. Radar altimeter echoes can even be used to discriminate between open water and thin sea ice ($< 25$ cm) in leads and may allow direct estimation of thin ice thickness (Müller et al., 2023). The specular reflection of a lead surface in nadir/zenith direction for open water or young sheet ice also dominates the radar echo if it only covers 1% of the illuminated area (Drinkwater, 1991). Radar echoes over sea ice surfaces are notably weaker and the return echoes are distributed over a larger time window due to diffuse scattering, a wider surface height distribution per footprint, and partly backscattering snow layer. The specular versus diffuse backscatter mechanisms of leads and sea ice respectively also result in a strong over-representation of the lead area fraction within a radar altimeter footprint for mixed surface types. Any binary lead/ice classification will therefore result in a higher lead detection rate than the true lead area fraction in the absence of misclassifications. Müller et al. (2023), however also show that surface-type classification algorithms may not correctly label radar echoes as leads in the presence of thin ice, thus potentially reducing the lead count. We keep these competing and not quantifiable biases in mind for the interpretation of the $LF_{CS2}$.



Several waveform parameters have been proposed for surface type classification, but the fundamental concept remains un-
changed to the earliest studies of sea surface height (Peacock, 2004) and thickness (Laxon et al., 2003) in the Arctic Ocean. In
the "Level-3 gridded sea-ice thickness and auxiliary parameters" product, the surface type classification scheme based on (Paul
et al., 2018) attributes each radar waveform into three categories: lead, sea ice, and ambiguous. The thin ice class proposed in
Müller et al. (2023) is not included. The surface type, among other geophysical parameters, is then gridded from the original
along-track resolution of approximately 300 meters for weekly and monthly periods onto an EASE2 grid with a spatial reso-
lution of 25 km. To improve the comparability with the other methods in this study, we create a custom gridded product with a
temporal resolution of one day and a spatial resolution of 12.5 km respectively. We use the python package pysiral (Hendricks
et al., 2021b), for the surface type classification and geophysical retrieval using the CryoSat-2 sensor as well as for the custom
gridding.

The gridded files contain the number of waveforms classified as either lead or sea ice (variable "stat_n_valid_waveform") and
the fraction of lead detections (variable "stat_lead_fraction"), the $LF_{CS2}$. Multiplying the two parameters yields the absolute
number of lead detections (C2 absolute) per grid cell area and period. We use C2 absolute as a secondary metric because the
$LF_{CS2}$ can have a strong sampling bias if only a few waveforms are detected per grid cell.

## 3 Evaluation of lead fractions based on divergence during MOSAiC

Our aim is to present and evaluate our lead fractions based on SAR-derived divergence along the MOSAiC drift track. First,
we analyze the properties of the accumulated divergence-based lead fractions ($LF_{accu.\ div}$) on different spatial scales (Section
3.1). Second, we present the LKF-derived lead fractions $LF_{LKF\ pixel}$ and $LF_{LKF\ fraction}$ (Section 3.2).

### 3.1 Accumulated divergence-derived lead fractions

Figure 2 displays snapshots of lead openings from March 20 to March 27. Advected to and plotted on a SAR image from March
27, the different colors show accurately where and when leads opened in the past 7 days (Figure 2e). Most of those leads were
not closed dynamically and can still be identified by eye based on their lower radar backscatter on the SAR image. The leads
follow a preferred direction roughly perpendicular to the sea ice drift. Despite their spatial proximity, the deformation history
of the leads differs considerably. Some of them opened up and closed several times while others opened up only once. Next,
we give a detailed example of the deformation history of a lead shown in Figure 3.

### 3.1.1 Deformation history of a single lead

Figure 3 displays the temporal evolution of a lead between March 15 and March 27 at near-daily resolution. Prior to March
15, the ice pack was closed, but bright lines in the SAR backscatter hint at previous deformation events. The lead experienced
opening (March 16-18), closing (March 18-23), and re-activation (March 25-26). Please note, the low temporal coverage
prevents us to analyse any sub-daily deformation due to e.g. tides. We calculated the width of that lead from the lead fractions
assuming that all the divergence resulted in opening in one direction. We evaluated the lead width estimates against visually




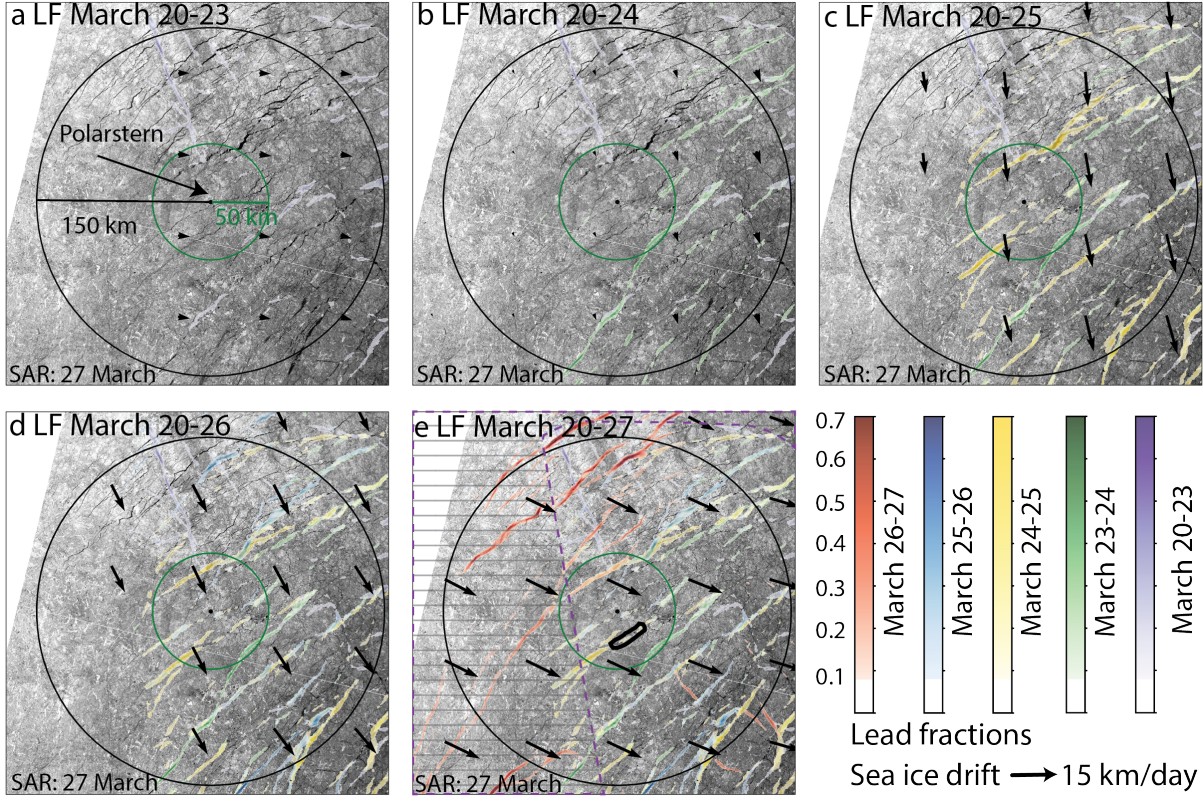

**Figure 2.** Example of accumulated divergence-derived lead fractions (LF$_{accu. div}$) from March 20 to March 27. The Sentinel-1 image from March 27 is overlaid by advected lead fractions from different time instances. The colors indicate the timing and magnitude of the lead opening. The black arrows show the sea ice velocity of the latest time step, e.g. e: Mach 26/27. The black and green circles around the position of R/V *Polarstern* have a radius of 150 km and 50 km, respectively. The non-dashed area in panel e indicates where lead fractions from all 7 days are available. The deformation history of the feature encircled by the thick, black line within the green circle is shown in Figure 3.

detected changes in the backscatter of the SAR image. Our estimates correspond well to manually measured widths of 1.8-2.0 km and 1.2-1.5 km on March 19 and March 27, respectively. On those two days, the presence of frost flowers on the lead ice turned the leads into a highly diffuse scattering surface with a strong backscatter contrast to the surrounding ice. We do not provide estimates for the other days as wind-affected new ice formation and finger-rafting quickly transformed the smooth lead ice into a rough surface whose SAR backscatter signal was indistinguishable from the surrounding ice.

This example provides valuable insights into the life cycle of leads and the capability of the accumulated lead fraction product to resolve the different phases of opening, closing, and reactivation. As expected from modeling studies(e.g., Wilchinsky and Feltham, 2011), divergence and convergence on consecutive days were concentrated on the same ice that was already weakened by previous deformation. The example also demonstrates that only when accumulating lead fractions over several days, we





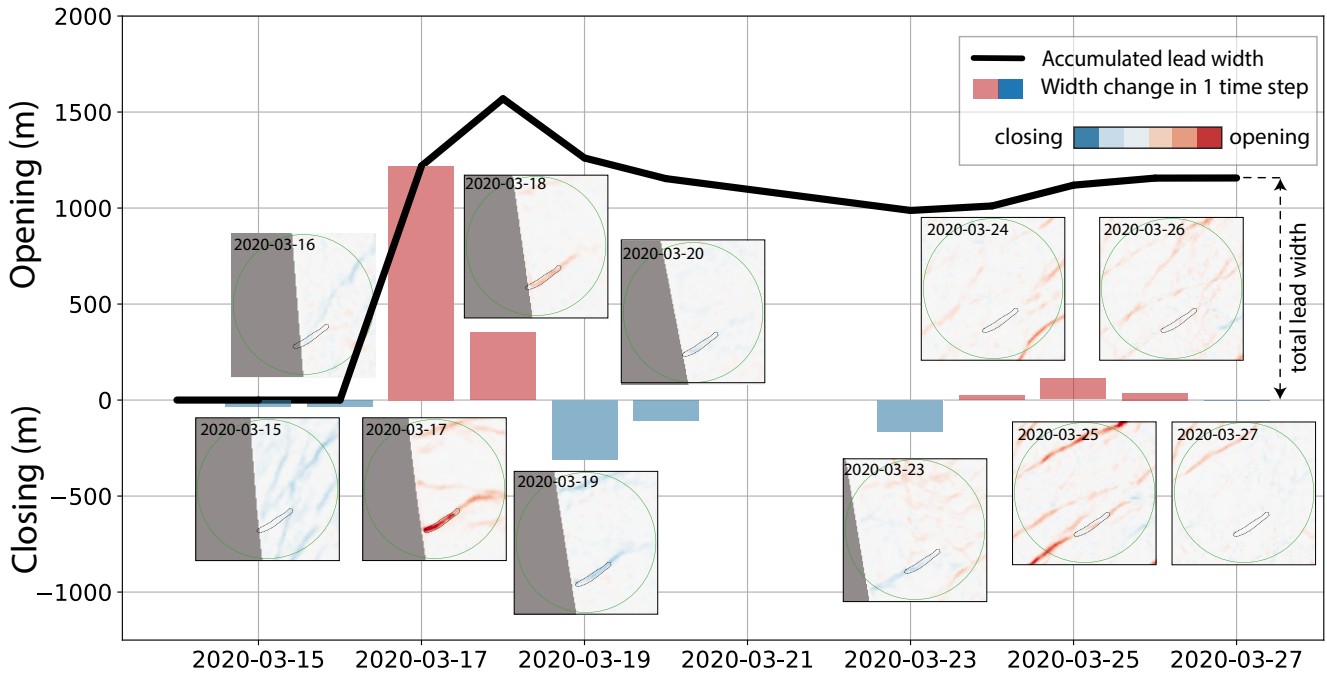

**Figure 3.** Time series of lead opening, closing, and reactivation from March 14-27. The aerial plots show opening (red) and closing (blue) in a circle with 50 km radius around R/V *Polarstern*. The values of the pixels within the dashed line were averaged and are shown by the blue bars. The black dashed line connecting the blue bars accumulates the opening and closing given by the blue bars. The lead width was calculated from the lead fractions assuming that divergence took place only in one direction.

resolve the thin ice present on March 27. Combined with a thermodynamic growth model, one could calculate the thickness of
the thin ice at any time step (see, e.g., Kwok and Cunningham, 2002; Kwok, 2006).

### 3.1.2 Statistical analysis of lead lifetime, reactivation, and lead width

The accumulated divergence lead fractions provide valuable information for conducting statistical analyses on various properties of leads, including lead lifetime, reactivation percentage, and lead width. In the following, we have analyzed and summarized those properties for all leads in our dataset.

Lead lifetime characterizes the duration a certain lead was open within the 10-time step study period. Figure 4 displays the time series of lead lifetimes. The lifetime follows an exponential fit (Figure 4b, left y-axis) with reduced residuals when limited to the range of 3-11 days. Analyzing the accumulated relative frequency (Figure 4b, right y-axis), we observe that the most common lifetime is 2 days, accounting for 33% of the leads. This coincidence with the shortest lifetime we can resolve, i.e., the actual most frequent lifetime might be shorter. The median lifetime is 3 days, indicating that 50% of the leads close
within this timeframe. After 5 days, 72% of the leads have closed, while only 2% remain open longer than 11 days. Phases





of reduced lifetime often coincidence with strong convergence events (green shading in Figure 4a), while there is no link for others (orange shading in Figure 4a). The large standard deviation of each time step indicates additional spatial variability despite being influenced by similar large-scale wind and ocean forcing.

Reactivation refers to the reopening of a previously closed lead. Over the whole time series, on average 9.6±4.6% of the leads
were reactivated within the 10-time step time period. The reactivation percentage can reach up to 24% in March, as shown in Figure 4a. The temporal variability of reactivation closely follows the one of the lead lifetime (Person R=0.39).

Lead width, as defined in the previous section, is depicted in Figure 5. The smallest lead width we can detect must exceed the uncertainty of the lead fraction, i.e., 56-112,m (see Section 3.1.3). We have chosen 56,m as the lower limit. As expected, the lead width distribution follows a power law (exponential fit in 5b) and exhibits a "heavy tail" with lead widths of up to 1500,m.
Beyond 1400 m the exponential relationship breaks down which is why we define 1400 m (2 pixels of the divergence product) as the upper detection limit of our algorithm. Nevertheless, leads wider than 1.4 km can still be resolved in our data product by summing up several pixels along the opening direction because their actual width is smeared out over several pixels.
We observe a seasonal variability in lead width, with larger leads and greater spatial variability in spring (Figure 5a). Combining lead width and the number of pixels with leads provides also insights into the variability of the localization of the deformation,
i.e., whether there are many small leads or a few large ones. The variability of lead width and the number of pixels with leads appears to be correlated (Pearson R=0.26) with more lead pixels when there are larger leads, which is particularly evident in October and November (Figure 5a).

In conclusion, the dataset presents a valuable resource for studying the physical and mechanical properties of leads larger than 56-112 m. Combined with ice pack properties, forcing fields, and a thermodynamic growth model, a detailed analysis of the
dataset can reveal an in-depth process understanding of sea ice mechanics and their role in the sea ice mass balance.

### 3.1.3 Uncertainties of the accumulated divergence-derived lead fractions

We identify two main sources of uncertainty accumulated divergence lead fractions: (1) Uncertainty related to the advection scheme and (2) uncertainty of the lead fraction magnitude. In addition, there are lower limits for the lead lifetime and lead width due to the temporal and spatial sampling limitations of the dataset.

The uncertainty of the advection scheme originates from errors in the drift calculations. Hollands and Dierking (2011) state a tracking error of ±0.8–1.6 pixels, i.e. ±40–80 m. Assuming a homogeneous drift field, the tracking error accumulates to a maximum of 400–800 m, i.e. one grid cell of the lead fraction product. For a strongly heterogeneous drift field, von Albedyll et al. (2021a) estimated an accumulated tracking error of 1200 m for 10 time instances using the same drift algorithm (see Figure 1 in supplementary material). Thus, the uncertainty of the advection scheme is small and in the order of 1–2 pixels. In
the spatial plots, e.g., Figure 3, we can confirm the high spatial accuracy. The deformation zone stays concentrated in a narrow zone without any signs of significant "smearing out" due to possible discrepancies in the advection scheme.

We calculate the uncertainty of the lead fraction magnitude of a single time step from the tracking uncertainty using error propagation assuming no geolocation errors following Dierking et al. (2020). Adapting their equation 17, the uncertainty of

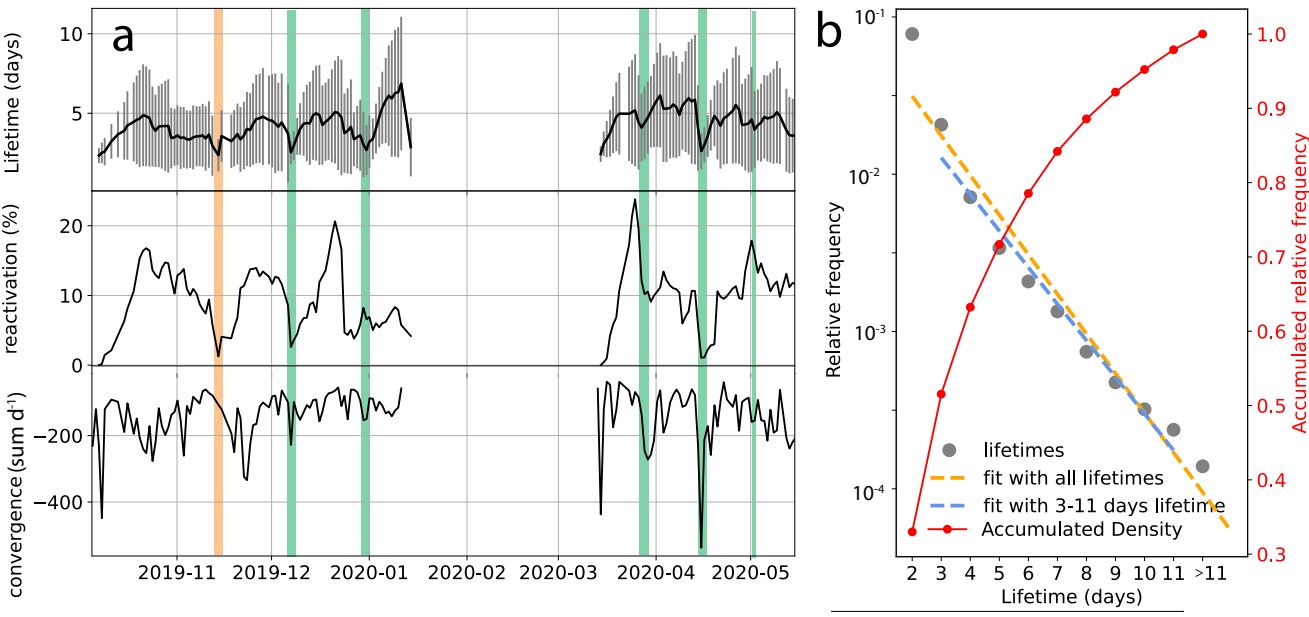

**Figure 4.** Time series (a) and distribution (b) of lead lifetime based on the complete dataset. Panel a displays the time series of the mean (+standard deviation) lead lifetime of each day, reactivation percentage, and spatially summed-up convergence. Green shading highlights convergence events with an impact on the lifetime while orange shading highlights reductions in lifetime without corresponding strong convergence events. Panel b shows the logarithmic distribution of all lead lifetimes (gray, left y-axis) and two exponential fits (blue, orange, left y-axis). The right y-axis shows the accumulated density distribution of the lifetimes.

the lead fractions $\sigma_{LF}$ is given by the ratio of the tracking uncertainty $\sigma_{tr}$ and the temporal $\Delta T$ and spatial $L$ scales:

$$\sigma_{LF} = \frac{\sqrt{2}\sigma_{tr}}{\Delta T L} \tag{2}$$

With a tracking uncertainty of $\sigma_{tr} = 40-80$ m (Hollands and Dierking, 2011), a spatial scale of $\Delta L$=700 m, and a typical time scale of $\Delta T$=1 day, this results in an uncertainty of $\sigma_{LF}$= 0.06-0.16 for a single lead fraction pixel per day. Translated into lead width, this corresponds to 56–112 m per day when assuming that the lead has opened only along one dimension. Averaging over larger spatial scales and using the adjusted tracking uncertainty of $\sigma_{tr}$=1200 m for 10 days, we calculate the uncertainties of the averaged lead fractions assuming independent error propagation with:

$$\sigma_{LF} = \frac{\sqrt{2}\sigma_{tr}}{\Delta T L \sqrt{(n)}} = \frac{\sqrt{2}\cdot 1200}{10\cdot 700\cdot\sqrt{(n)}} \tag{3}$$

where $n$ is the number of pixels that fit into circles with radius 10 km, 50 km, 100 km, and 150 km. This calculation yields uncertainties for the averaged lead fractions of 0.0096 (10 km), 0.0019 (50 km), 0.00096 (100 km) and 0.00064 (150 km).

As uncertainties grow with more accumulation steps, we explore the upper and lower limits of the number of accumulation steps. In winter, thermodynamic growth sets an upper limit on the accumulation time instances. Depending on the heat fluxes





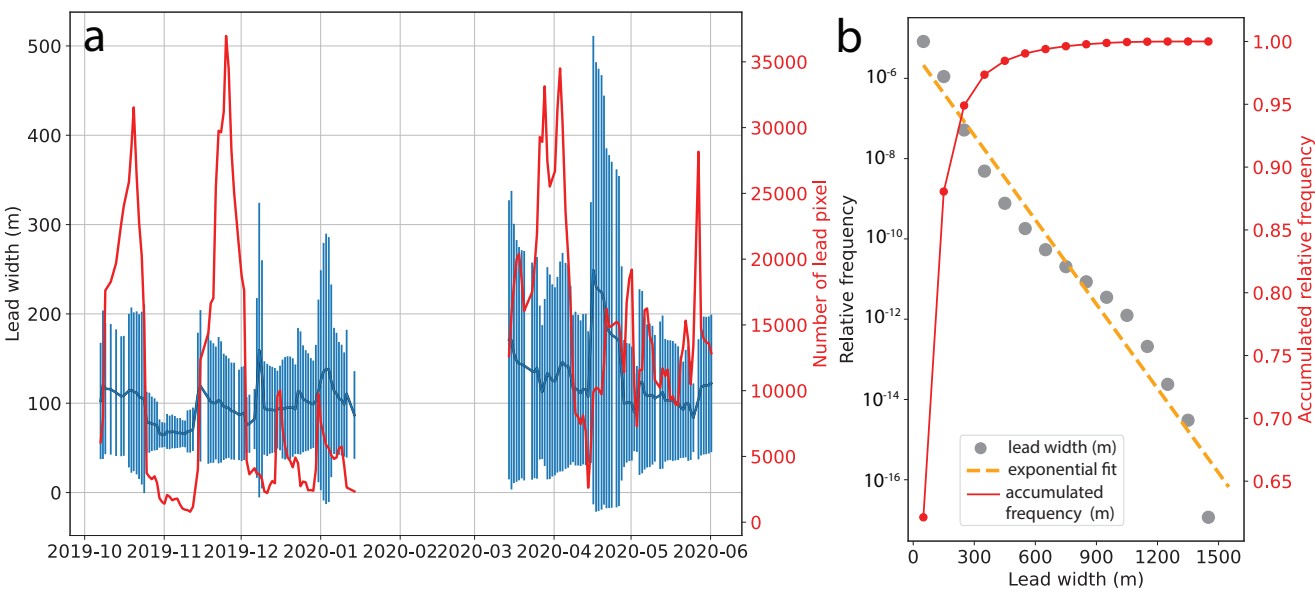

**Figure 5.** Time series (a) and distribution (b) of lead width based on the complete dataset. Panel a displays the time series of the mean (+standard deviation) lead width of each day (blue, left y-axis) and the number of pixels showing leads (red, right y-axis). Panel b shows the logarithmic distribution of all lead widths (gray, left y-axis) and the exponential fit (orange, left y-axis). The right y-axis (red) shows the accumulated density distribution of the lead widths.

and the threshold for lead ice thickness set by the research question, a lead is "thermodynamically closed" after several days. We chose 10 time instances corresponding to at least 10 days as an upper limit. Under the typical (winter) growth conditions during MOSAiC, lead ice thickness reached on average more than 30 cm (min:1 cm, max:50 cm) after 10 days (Nicolaus et al., 2022) which is thicker than most thresholds for lead ice. The dynamic lifetime of the leads sets the lower limit of the required

accumulation steps. We compared the time series with zero to 10 accumulation steps and found that already 5 accumulation steps can explain roughly three-quarters of the magnitude and variability of the "full", 10 accu. div time series. This fits well with the observation that 72% of the leads are closed after 5 days (see Figure 4b, Section 3.1.2). Hence, we suggest using at least 5 time instances ($LF_{5x \text{ accu. div}}$) to describe the temporal evolution of the leads.

Lead width and lifetime are known to follow a power law also well beyond the presented range of values (e.g., Hutter et al.,

2019; Thielke et al., in review, see also Figures 4, 5). In other words, there are many unresolved, small leads (<56 m) with short lifetimes (< 2 days). In addition, this dataset does not resolve the true lifetime and width of those 2% of the leads that stayed open for more than 11 days. Resolving all those leads would directly affect the mean properties.

The uncertainty of lead fraction magnitude varies with numerous effects like the backscatter contrast or the orientation of the grid to the deformation zone (Bouillon and Rampal, 2015; Griebel and Dierking, 2018). However, the lack of high-resolution

reference datasets for comparisons (see Section 4) with our results restricts us to this theoretical approach that provides an

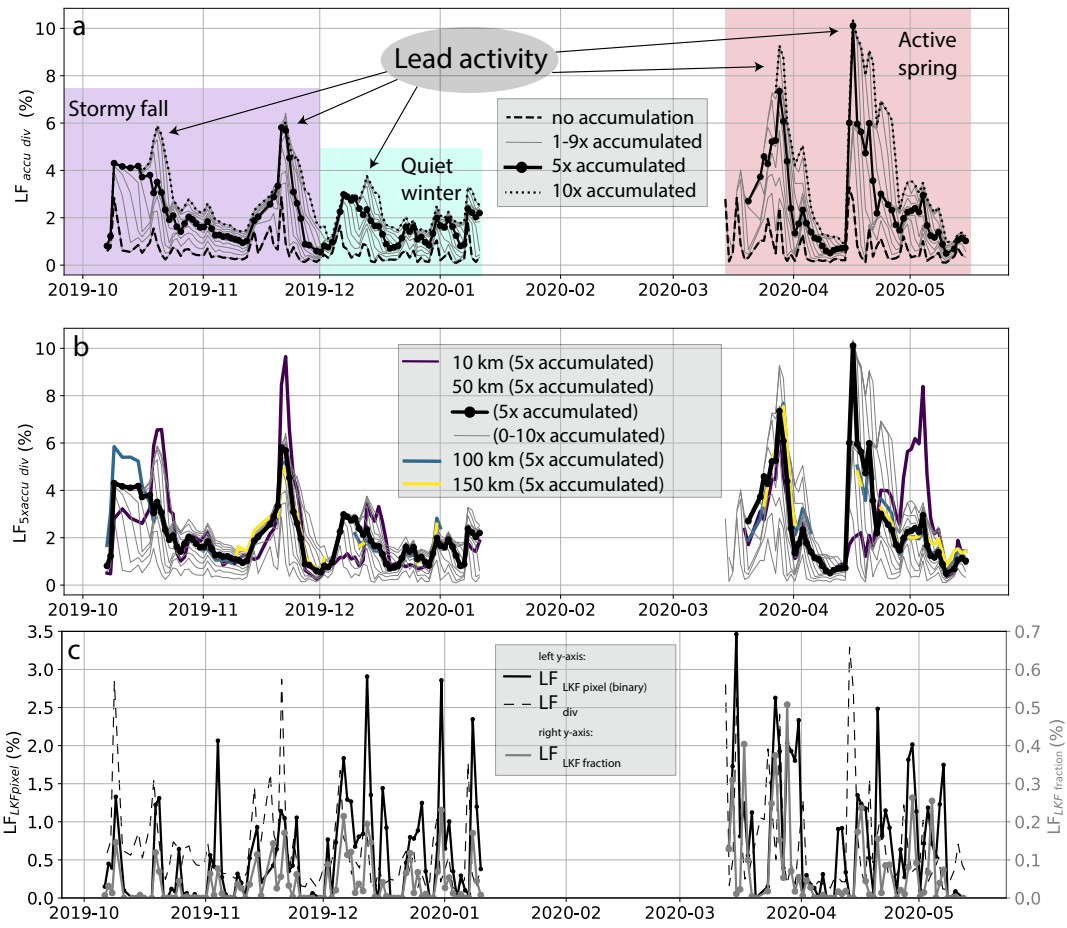

**Figure 6. LF$_{accu. div}$ and LF$_{LKF}$ time series**. Panel a shows accumulated LF$_{accu. div}$ on the 50 km scale with a varying number of accumulation time instances. The time series with no accumulation (dashed), 5x accumulated (bold), and 9x accumulated (dotted) are highlighted. The three main phases and several periods of higher lead activity are marked with colored shading and arrows, respectively. Panel b compares accumulated LF$_{accu. div}$ of different spatial scales. Panel c shows on a 50 km spatial scale the binary lead pixel number LF$_{LKF\ pixel}$ and LF$_{div}$ on the left y-axis and LF$_{LKF\ fraction}$ lead fractions on the right y-axis (mind the different scales).

upper limit. Overall, we conclude that we accurately record when and where a lead opened and closed for up to 10 time instances, but the actual lead width is more uncertain.

### 3.1.4 Time series of accumulated divergence-derived lead fractions during MOSAiC

The mean (accumulated) LF$_{accu. div}$ range between 0.61% (LF$_{div}$, no accumulation) and 3.2% (LF$_{10x\ accu. div}$, 10x accumulated)
with a maximum of 3.3% (LF$_{div}$) and 10.3% (LF$_{10x\ accu. div}$), respectively. Figure 6a displays the time series of (accumulated)





$LF_{div}$ for different numbers of accumulation steps, where $LF_{5x\ accu.\ div}$ is highlighted in bold. In the following, we will focus on the $LF_{5x\ accu.\ div}$ time series.

The $LF_{5x\ accu.\ div}$ time series is roughly split into three phases: a stormy fall (Oct-Nov), a quiet winter (Dec-Jan), and an active spring (Mar-May, Figure 6a). Those phases align with the general seasonality of the dynamic regime during the MOSAiC drift
(personal communication, A. Bliss, J. Hutchings, see also, e.g., Krumpen et al., 2021; von Albedyll et al., 2022, for sea ice dynamics along the drift). Within the three main phases, there are several peaks of lead activity lasting 1-1.5 weeks (arrows in Figure 6a). The highest lead activity, characterized by multiple events of a duration of 1-2 days with high lead fractions, was reached in March 2020, when R/V *Polarstern* was in the Western Nansen Basin, a region that is generally characterized by high lead fractions (Figure 15 in Krumpen et al., 2021). Interestingly, the frequency of lead events stayed roughly constant during
the seasons, but their larger magnitude and persistence in fall and spring created the offset in the accumulated lead fractions. When interpreting this time series, one needs to keep in mind thermodynamic growth quickly covering leads with new ice.

The three phases correspond well with the atmospheric forcing and the consolidation state of the ice pack. In the fall when R/V *Polarstern* was located in the Siberian Arctic, the sea ice was still freezing-up and was hit by an elevated number of cyclones in November (Rinke et al., 2021; Nicolaus et al., 2022; von Albedyll et al., 2022). In the quiet winter phase in the Central Arctic
(here only documented until mid of January), the ice pack consolidated completely with only one cyclone passing through in December. In the active spring, cyclone activity increased again and a sequence of storms first broke and then easily deformed the ice pack (Rinke et al., 2021; Nicolaus et al., 2022). This increase in lead activity corresponds well to R/V *Polarstern* approaching the Western Nansen Basin, a region that is generally characterized by higher lead fractions (Figure 15 in Krumpen et al., 2021).

### 3.1.5 Accumulated divergence-derived lead fractions on different spatial scales during MOSAiC

Panel b of Figure 6 compares $LF_{accu.\ div}$ of different spatial scales, described by circles with radii from 10 km to 150 km centered around R/V *Polarstern*. Lead fractions of the different scales are generally similar in magnitude and temporal evolution. They differ less than the different accumulation time instances. This means that the deformation was more consistent on spatial scales up to 150 km than persistent on temporal scales > 5 days. However, on the smallest spatial scale of 10 km, we note some clear
deviations from the overall pattern. Here, the localized and intermittent nature of deformation starts to become apparent. Due to the lower data coverage, the time series of 100 km and 150 km are incomplete as we only consider points with a coverage > 50%. We conclude that the 50 km time series is representative of the larger surroundings of the MOSAiC central observatory based on a high correlation of >0.9 with the 100 km and 150 km time series.

### 3.2 LKF-derived lead fractions

Panel c of Figure 6 presents the LKF-derived lead fractions. The binary lead pixel number $LF_{LKF\ pixel}$ is displayed together with the native product $LF_{div}$ on the left y-axis, and the LKF lead fractions $LF_{LKF\ fraction}$ are shown on the right y-axis (different scale). $LF_{LKF\ pixel}$ have a similar magnitude as the $LF_{div}$ with a mean of 0.65% and a maximum of 3.46% (on March 15, 2020).



In contrast and as expected from the processing, the average of the $LF_{\text{LKF fraction}}$ are two magnitudes smaller than LFdiv with a mean of 0.05% and a maximum of 0.51% (on March 28, 2020). Both time series exhibit a very similar temporal variability

which is why we summarize them together as $LF_{\text{LKF}}$. Because $LF_{\text{div}}$ and the $LF_{\text{LKF}}$ are based on the same divergence fields, the observed, very similar temporal variability (Pearson R=0.78) is expected. In contrast to the $LF_{\text{div}}$ time series, lead events in the $LF_{\text{LKF}}$ time series are more prominent, clear signals in the time series with otherwise mostly zero values.

The observed differences align well with the differences in the retrieval techniques. The LKF detection algorithm of the $LF_{\text{LKF}}$ functions as a strict filter on $LF_{\text{div}}$ by reducing unwanted noise and highlighting strong events. However, it also removes most

likely real divergence events that are either not strong enough or do not result in elongated shapes. The agreement between the magnitudes obtained from $LF_{\text{LKF pixel}}$ and $LF_{\text{div}}$ results in several conclusions. By assigning a lead fraction of 1 to every LKF pixel, we end up overestimating the lead fraction. This happens because only 0.4% of all lead pixels contain leads wider than 700 meters (as depicted in Figure 5). Yet, this overestimation is counteracted by the morphological thinning of leads into 1-D features. Both effects seem to compensate for each other on average. Therefore, we conclude that $LF_{\text{LKF pixel}}$ are a first guess

to concentrate the spread-out lead signal in the divergence data into a single pixel. It is important to note that, unlike $LF_{\text{accu. div}}$, $LF_{\text{LKF}}$ only consider newly formed leads.

## 4   Comparison of different lead products

This section compares the results of our two new lead products with results from the 6 existing lead products described in Section 2.3. We compare the time series with respect to their mean values (Section 4.1), their temporal variability (Section

4.2), and their ability to resolve leads spatially (Section 4.3).

We restricted our analysis to the time period from October 5, 2019, to May 15, 2020. During the subsequent melt season, most of the retrieval methods suffer from large uncertainties and are thus not available. For the temporal comparison, we compared mean lead fractions in a circle with 50 km radius around the position of R/V *Polarstern* at the acquisition time of the SAR images. We chose this scale because it is representative of the wider surroundings (see Section 3.1.5), captures the LKF

structures, comprises the extended network of measurements conducted during MOSAiC, and is a compromise between the different resolutions and coverage of the various lead products. For the time scale of the accumulated lead fractions, we have chosen $LF_{\text{5x accu. div}}$ (see Section 3.1.3).

### 4.1   Mean and variability of lead fractions in different products

Table 1 provides a summary of the average lead fractions from various time series along with their respective variability

measures, represented by the standard deviation and the coefficient of variability. The coefficient of variability is calculated as the ratio of the standard deviation to the mean value, serving as an indicator of the relative variability for each set of data.

The mean lead fractions among the lead products vary by two magnitudes between 0.05 ($LF_{\text{LKF}}$) and 9.22% ($LF_{\text{CS2}}$) while their variability is very similar with coefficients of variability between 0.71-2.04. For a plausibility check of the magnitude of the lead fractions, we compare the mean lead fractions to mean open-water fractions derived from airborne ice thickness





**Table 1. Average properties of different lead products in a 50-km circle around R/V _Polarstern_.** The mean lead fraction and its standard deviation are given with the coefficient of variations defined as the ratio of the two former quantities. C2 absolute is given in absolute numbers, not percentages. The length of the time series varied due to the lack of satellite coverage or data gaps, e.g., due to clouds.

| Dataset | Mean fraction (%) | Standard deviation | Coefficient of variation | Length of time series (days) |
|---|---|---|---|---|
| $LF_{div}$ | 0.61 | 0.62 | 1.0 | 147 |
| $LF_{5x\ accu.\ div}$ | 2.23 | 1.57 | 0.71 | 143 |
| $LF_{LKF\ fraction}$ | 0.05 | 0.09 | 1.64 | 147 |
| $LF_{LKF\ pixel}$ | 0.65 | 0.74 | 1.12 | 147 |
| $LF_{classified\_SAR}$ | 0.60 | 1.23 | 2.04 | 149 |
| $LF_{MODIS}$ | 5.77 | 6.52 | 1.13 | 117 |
| $LF_{Heli\_TIR}$ | 1.16 | 1.07 | 0.93 | 9 |
| $LF_{PMW}$ | 5.08 | 3.61 | 0.71 | 224 |
| $LF_{CS2}$ | 9.22 | 10.30 | 1.12 | 185 |
| C2 absolute | 2.38 | 2.25 | 0.94 | 185 |

observations during MOSAiC reported in von Albedyll et al. (2022). Because the open-water fraction of the EM ice-thickness measurements is limited to 0–10 cm thick ice, it approximates the fraction of leads that opened up on that particular day or slightly before in the freezing season. Taking the average of all nine surveys between October 2019 and April 2020 gives an open-water fraction between 0.02% and 0.81% with a mean of 0.35%. This number most likely underestimates the true open-water fraction due to the footprint averaging of the EM airborne ice thickness measurements. Nevertheless, it provides a rough

estimate in the same order as the mean $LF_{div}$, $LF_{classified\_SAR}$, and $LF_{Heli\_TIR}$.

The observed differences in magnitude provide clear evidence for the major differences in the "lead definition" (Section 2.1) of each retrieval method, but also in their spatial and temporal coverage and resolution. Since the variability is similar, we suggest that all retrieval methods react similarly sensitive to changes in the real lead fractions, but with different magnitudes. $LF_{5x\ accu.\ div}$, $LF_{MODIS}$ and $LF_{PMW}$ overestimate the open-water fractions compared to the EM thickness observations. This

fits well with the assumption that their lead fractions also include thin ice. However, the $LF_{Heli\_TIR}$ are based on the same measurement principle as $LF_{MODIS}$, but have substantially smaller values. Therefore, other factors such as spatial resolution, spatial coverage, and atmospheric conditions also play a role. $LF_{div}$ and $LF_{classified\_SAR}$ detect primarily open-water leads which is supported by their reasonably good fit to the observed EM open-water fractions.

## 4.2 Temporal variability of different lead products

After the mean fractions, we compare (1) the temporal variability and (2) the temporal resolution and coverage of our lead products based on divergence with other lead products. Figure 7 presents the different lead-fraction time series with different y-axes.



**Figure 7. Time series of lead fractions from different products.** Colored and labeled (a-c) bars indicate common events. Red shading (a1-a6) highlights lead events in the divergence-based products and most other sensors. The yellow shading (b1-b2) shows events visible in the $LF_{CS2}$ and the $LF_{div}$ and $LF_{LKF}$ time series. The blue shading (c) highlights the strongest lead event in the $LF_{div}$ and $LF_{accu. div}$. The scale of the different y-axes differs substantially.



**(1) Common variability of the lead-fraction time series**

The seasonal variability of the $LF_{classified\_SAR}$ time series is similar to the one of the $LF_{div}$ and the $LF_{LKF}$ with more active
phases in fall and spring. The time series of $LF_{classified\_SAR}$ has several active phases or individual events in common with
the divergence-based products which are marked in red in Figure 7 and labeled a1-a6. For the first, very active phase in
October (a1) there is no one-to-one correspondence between the individual lead events, but all available datasets, including
the $LF_{CS2}$ indicate the presence of several leads. Between November and March (a2-a6), the $LF_{classified\_SAR}$ agree with the
most pronounced lead opening events in $LF_{div}$ and $LF_{LKF}$ that correspond in most cases also to maxima in the accumulated
$LF_{5x\ accu.\ div}$. Smaller events that appear exclusively in $LF_{div}$ and $LF_{LKF}$ were not consistently identified by $LF_{classified\_SAR}$. This
suggests that the divergence-based products are more adept at capturing lead events than $LF_{classified\_SAR}$, although they may also
contain some noise.

There are also pronounced differences between $LF_{classified\_SAR}$ and the divergence-based lead fractions. For example on April
15-16, 2020, the strongest lead event of $LF_{div}$ (blue shading, c) lacks a counterpart in the $LF_{classified\_SAR}$. In fact, this peak
corresponds to a large shear zone in the study area creating open water and ice rubble.

It is interesting to note, that the duration of the individual events lasts typically 1-2 days similar to $LF_{div}$ and $LF_{LKF}$ while the
lower-frequency variability of $LF_{accu.\ div}$ indicates that some of the leads stayed open longer than a day. We thus conclude that
$LF_{classified\_SAR}$ detects predominantly open water and only to a minor degree leads covered by thin ice.

Frequent data gaps in the $LF_{MODIS}$ time series due to clouds complicate a comprehensive comparison. The seasonal evolution
agrees with the ones of the other time series with high lead fractions in March 2020. One major event in December 2019
(yellow shading, b1) is shared with $LF_{div}$, $LF_{LKF}$, and $LF_{CS2}$ but has a much larger amplitude in $LF_{MODIS}$.

The $LF_{Heli\_TIR}$ time series consists only of nine temporal snapshots which prevent an in-depth interpretation of the variability.
In addition, the helicopter-borne dataset is very limited in space. Overall, the trend, suggested by the few data points, towards
higher fractions in spring aligns well with the other lead-fraction time series.

The $LF_{PMW}$ time series shows a gradual decline in lead fraction from fall to spring, a trend not observed in the other time series.
Notably, for the first time, zero lead fractions are recorded around mid-March, coinciding with a shift in the general pattern,
leading to the emergence of more distinct events. Among these events, three coincide with events observed in other products
(a5-a6). It is important to mention that during the strong shearing event on April 15-16, 2020, the ice concentration (shown on
the right y-axis in Figure 7) decreases, while the lead fraction remains small. Based on our observations, we propose that the
gradual decrease in LFPMW primarily reflects the thermodynamic thickening of thin ice rather than a change in the presence of
leads. However, starting from mid-March, LFPMW appears to predominantly capture thin ice formed in refrozen leads while
being less sensitive to open water, as evidenced by the April 15-16 event. Please note, the strong decrease in concentration after
April 16 is related to the effect of glazing on the retrieval algorithms due to a warm air intrusion (Rückert et al., in review).

The seasonality of the $LF_{CS2}$ time series is slightly different from the divergence-based lead products with the highest lead
activity in fall and only a few events recorded in spring. With this, it corresponds better to $LF_{classified\_SAR}$. Potential causes of
this deviation could arise from the abundant thin ice in the fall that could have been classified as leads by the retrieval algorithm.



Similar to LF$_{LKF}$, the LF$_{CS2}$ time series consists of events that last normally 1-3 days that can be easily separated from each other. There is good agreement between the LF$_{CS2}$ and several other products for several events (a1, a2, a5). In addition, they agree with LF$_{div}$ and LF$_{LKF}$ on additional events in December 2019 (yellow shading, b1-b2). The LF$_{CS2}$ suffer from a low spatial coverage that likely causes a sampling bias in the lead indication. For example, the ice affected by the major lead event mid of April (blue shading, c) was not covered by the swath of the satellite.

Overall, for shorter periods than the seasonal cycle, there is only anecdotal agreement between the different lead product time series. This lack of general similarity strongly complicates establishing some kind of "common ground" for the evaluation of lead products.

**(2) Temporal and spatial coverage**

Lastly, we compare the temporal resolution and coverage. Table 1 shows that LF$_{PMW}$ are the most complete time series, while LF$_{MODIS}$ have the least valid days. The divergence-based time series perform in the midfield, however, they suffer from a very irregular distribution of the gaps caused by the lack of satellite coverage north of approximately 87°N. Especially in those regions, lead fractions from other sensors than Sentinel-1 and most other SAR satellites are crucial. While those results are specific for the MOSAiC drift track, they still demonstrate well the limiting factors of the different time series that are either sensor specific (no coverage beyond a certain latitude) or due to the retrieval technique (clouds).

### 4.3 Spatial comparison

The previous sections were concerned with comparing mean values and temporal variability. Next, we analyze how well LF$_{div}$, LF$_{5x\ accu.\ div}$ and LF$_{LKF}$ reproduce the location and size of the same leads compared to a visual reference. To do so, we focus on two case studies in November 2019 and March 2020 with different dynamic regimes.

**Single deformation event - November 1st-2nd, 2019**

Between November 1st and 2nd, 2019, two leads opened in the previously closed ice pack (Figure 8b). An approximately 70 km long and 350 m wide lead opened 25 km south of R/V *Polarstern*. A smaller lead with 33 km length and a maximum width of 250 m opened 11 km to the west of the ship. Both leads were closed again on November 3, 2019. Extrapolating from the ice-thickness observations from October 14 onto November 1, 2019, the modal ice thickness of the surrounding ice was likely around 0.5 m±0.1 m (von Albedyll et al., 2022).

We manually estimated a lead fraction of 0.25% from the SAR image (Figure 8a). The LF$_{div}$ captured the larger lead very well. As the only lead product, the LF$_{div}$ partly also indicated the formation of the smaller lead. We tested whether the observed width of the lead and the integrated divergence values along the opening direction of the lead match. We found that the LF$_{div}$ result in a lead width of 300–350 m which agrees well with the observed 350 m. However, the LF$_{div}$ also indicate divergences at several other spots where we could not find any visual signs of open water. This results in a slight overestimation of the lead fraction (0.7%). Those spurious detections are removed in the LF$_{LKF}$ but at the price of also removing any sign of the smaller lead. Since the larger lead is concentrated into a quasi-one-dimensional structure that is only one pixel wide, the estimate of




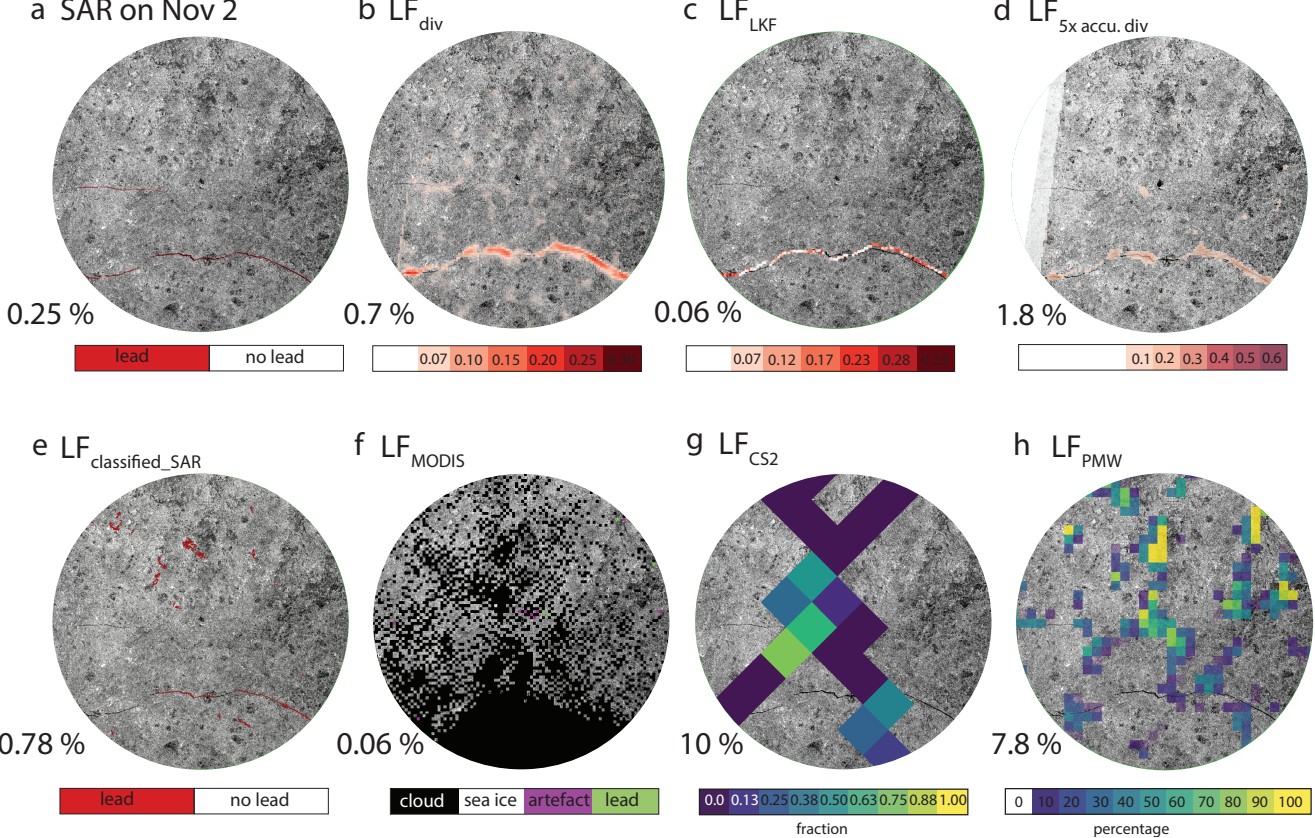

**Figure 8. Comparison of different lead products on November 2, 2019**. Panel (a) displays a SAR image of the ice pack on November 2, 2019, with two leads identified manually. Panel (b) displays the advected $LF_{div}$. Panel (c) shows the advected $LF_{LKF\ fraction}$. Panel (d) shows the advected, 5x accumulated $LF_{5x\ accu.\ div}$. All other lead products are shown in the second row. The numbers at the bottom left of each panel indicate the lead fraction. All circles have a radius of 50 km and are centered on the position of R/V *Polarstern*.

the lead width reduces to about 200–240 m. Altogether, this results in only a small lead fraction of 0.06%. The accumulated lead fractions $LF_{5x\ accu.\ div}$ show a similar distribution to $LF_{div}$, as anticipated due to the closed ice pack on November 1, 2019. The accumulation of false detections results in a higher lead fraction of 1.8%.

The $LF_{div}$, $LF_{LKF}$, and $LF_{5x\ accu.\ div}$ perform similarly well as the $LF_{classified\_SAR}$ in detecting the location of leads. The $LF_{classified\_SAR}$ benefit from a high spatial resolution that is one order larger than the one of $LF_{div}$ and $LF_{LKF}$ and captures the large lead pre-
cisely. However, the used lead classification algorithm detects reliably only leads with a minimum width of about 200 meters – 5 pixels of the original Sentinel-1 SAR scenes. More narrow leads and parts of a larger lead are not always classified as open water. The $LF_{classified\_SAR}$ detects additional features that are not visually identified as leads, similar to $LF_{div}$. This leads to an overestimation of the lead fraction by 0.78% compared to the visual estimate. While $LF_{MODIS}$ suffer from heavy cloud coverage (lead fraction: 0.06%), the $LF_{PMW}$ (7.8%) show some features that are most likely associated with thin ice rather than





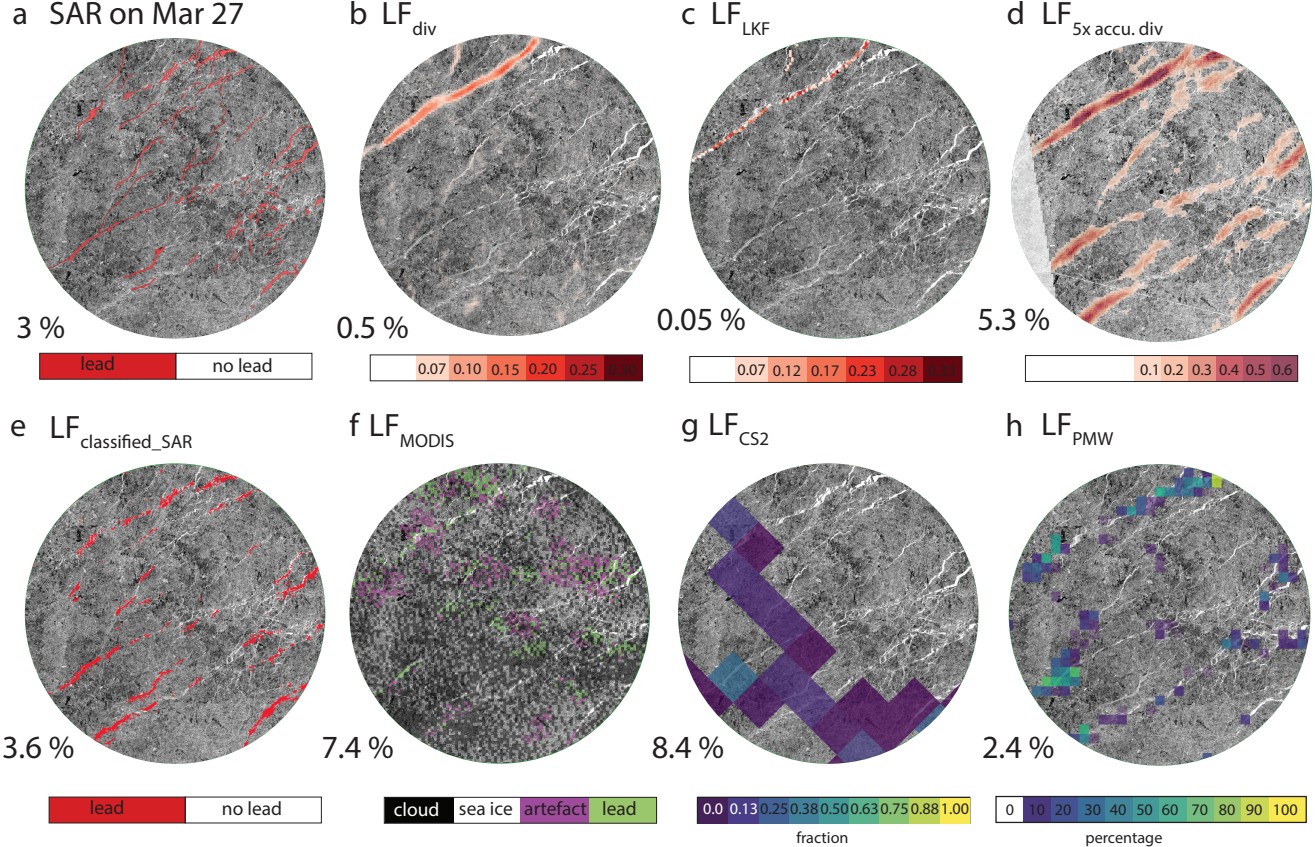

**Figure 9. Comparison of lead products on March 27, 2020**. Panel (a) displays a SAR image of the ice pack on March 27, 2020, with the largest leads identified manually. Panel (b) displays the advected $LF_{div}$. Panel (c) shows the advected $LF_{LKF\,fraction}$. Panel (d) shows the advected, 5x accumulated $LF_{5x\,accu.\,div}$. All other lead products are shown in the second row. The numbers at the bottom left of each panel indicate the lead fraction. All circles have a radius of 50 km and are centered on the position of R/V *Polarstern*.

leads. CryoSat-2 passed over the small lead and parts of the larger lead and captured higher lead fractions. The higher $LF_{CS2}$ of 10% confirmed the overall divergent drift regime that has opened probably a few additional smaller leads that are not visible on the SAR image.

**Dynamic phase with several leads opening - March 26th-27th, 2020**

March 26-28, 2020, was a very dynamic period with lead openings and closings. Several leads up to 1 km wide opened within 50 km distance to R/V *Polarstern*. Due to the low temperatures around –30 °C, the open water quickly refroze (Nicolaus et al., 2022; Shupe et al., 2022). The modal and mean ice thickness was around 1.7 m and 2.3 m, respectively (von Albedyll et al., 2022).



We manually estimated a lead fraction of 3% concentrating on the large lead systems (Figure 9a). Leaving out a few smaller
leads, this value probably still underestimates the true lead fraction. The $LF_{div}$ reproduce the opening of the leads in the upper
left part of the 50 km circle (Figure 9b). The respective divergence fields show that there was weak convergence along most
of the other lead locations. The fraction of 0.5% only reflects the newly formed leads. $LF_{5x\ accu.\ div}$ show that most of the leads
in the lower part of the circle had formed during the previous time instances (see also Figure 2) and suggests a higher lead
fraction of 5.3%. The visual estimate of 3% falls between the two products. This makes sense when considering the rapid new
ice formation in the leads up to 5 days old with the associated change in radar backscatter. For this case, the 2-day accumulation
with 3.3% comes closest to the visual estimate.

As $LF_{div}$, the $LF_{LKF}$ only indicate the active deformation zone in the upper left part of the study region and a lead fraction of
0.053%. Analog to the $LF_{div}$, the $LF_{LKF}$ showed a high fraction of 0.15% the day before on March 26, 2020. The $LF_{LKF}$ further
filtered out the weak convergence signals visible in the divergence lead fractions.

The $LF_{classified\_SAR}$ come closest to the manual estimate in magnitude (3.6%) and in location thanks to their high spatial res-
olution. In contrast to $LF_{5x\ accu.\ div}$, $LF_{classified\_SAR}$ do not identify some of the older leads, which is in line with the general
assumption that $LF_{classified\_SAR}$ resolves best leads with open water and smooth, thin ice. The $LF_{PMW}$ are with a fraction of
2.4 % still close to the visual estimate, but the analysis of the spatial distribution showed that the $LF_{PMW}$ product only resolves
parts of the leads (9h). Nevertheless, in contrast to the November case, the $LF_{PMW}$ significantly improved in predicting the
location of leads for the March case, as the few resolved structures were clearly aligned with leads seen on the SAR image
(Figure 9h). When not cloud-covered, $LF_{MODIS}$ detected individual leads, but not reliably. Interestingly, the $LF_{MODIS}$ of 7.4%
are still double the visual estimate. We speculate that there are two main reasons for this large estimate. First, where leads are
correctly identified, the fraction is overestimated due to the combination of gridding and the binary classification scheme (see
Section 2.3.2). Second, MODIS is detecting leads in some areas where leads were not observed visually, especially between
detected leads. This could hint at the presence of thinner leads that are not seen on the SAR image, warmer air around a lead
that "smears out" the lead signal, or old leads with thicker ice that is not seen by the other retrievals. The $LF_{CS2}$ indicate a lead
fraction of 8.4% which is high compared to the other sensors, but not a clear peak in the $LF_{CS2}$ (see also the November case
study). Even though the swaths indicate some coverage of the leads, the valid waveforms for the lead pixels are rather low. We
speculate that the smaller leads might have been missed since the data are not advected or filtered out because small leads often
create waveforms of mixed surface types that are removed intentionally in the processing.

We conclude that a spatial analysis of lead products, e.g., a visual comparison with higher-resolution optical or SAR data and
a plausibility check of the shape and stability of detected leads provides relevant information about the ability of products to
indicate leads on the chosen temporal and spatial resolution.





## 5  Discussion


The objective of this study was to analyze lead products based on divergence. We have calculated two lead products from the divergence: divergence-derived (accumulated) lead fractions ($LF_{div}$) and LKF-derived lead fractions ($LF_{LKF}$). Compared to the other lead datasets, we could identify four main advantages and disadvantages of the $LF_{div}$ and $LF_{LKF}$ that are outlined below. We note that our choice of comparison metrics might have overlooked other advantages and disadvantages of the lead products,

e.g. because they are present on a different spatial scale than the one we studied.

**Advantages of lead fractions based on divergence**

The first, and most important advantage of the $LF_{accu.\ div}$ is that combined with the drift information, they resolve the temporal evolution of individual leads. The knowledge of the deformation history of a lead enriches our understanding of the mechanical

properties of the ice. When combined with a thermodynamic growth model, it also allows reconstructing the thin ice part of the ice thickness distribution (e.g., Kwok and Cunningham, 2002). Statistics about lead lifetime, reactivation, and total lead width highlight the temporal and spatial variability of the deformation history of leads. Exploring this variability allows us to study changes in the mechanical properties related to ice pack properties across different regions and times and to compare them with sea ice models (e.g., Hutter et al., 2022; Ringeisen et al., accepted).

The second advantage of $LF_{accu.\ div}$ is the ability to detect small leads, and those which have SAR backscatter coefficients similar to ridges, e.g. caused by ice rubble or frost flower, while maintaining a large spatial coverage. Even though $LF_{classified\_SAR}$ indicate the location of leads with a 10 times higher spatial resolution, $LF_{div}$ were most reliable in resolving small leads (< 250 m) on the case study from November 2nd, 2019 (Section 4.3). Especially during predominantly shearing motion, when loose ice rubble is created which still allows for ocean-atmosphere exchange, the $LF_{accu.\ div}$ has a clear advantage over the

$LF_{classified\_SAR}$. While $LF_{CS2}$ most likely include even smaller leads, $LF_{div}$ have the advantage that they have a higher spatial resolution and larger spatial coverage than the gridded $LF_{CS2}$.

Third, $LF_{div}$ are easy to interpret as their magnitude is directly linked to the formation process of leads. The average magnitude of the $LF_{div}$ seems a realistic estimate of the open-water fraction compared to complementary, high-resolution airborne EM observations. The mean $LF_{accu.\ div}$ are larger because they include also leads covered by thin ice in addition to open water.

Combining the deformation history of $LF_{accu.\ div}$ with a thermodynamic growth model, would enable full control over the maximum allowed thickness in leads. In contrast, the larger $LF_{MODIS}$ and $LF_{PMW}$ classify thin ice up to a certain, unknown, ice thickness as leads.

Last, all of the time series based on SAR data ($LF_{div}$, $LF_{LKF}$, $LF_{classified\_SAR}$) have a high spatial resolution at moderate temporal coverage. The $LF_{LKF}$ indicate with similar precision as the $LF_{classified\_SAR}$ the location of the leads. While $LF_{PMW}$ and $LF_{CS2}$

have better temporal coverage (Table 1), they cannot compete with the spatial resolution or coverage of the SAR images, respectively. Surprisingly, in our study, $LF_{MODIS}$ fall behind the SAR time series with respect to the temporal coverage due to the high cloud coverage. In contrast, $LF_{MODIS}$ have the great advantage that with them we can cover the whole Arctic sub-daily without any gaps north of approximately 87°N. Thus, $LF_{MODIS}$ are well suited for long-term (months to years), pan-Arctic





studies of lead fraction trends.


**Disadvantages of lead fractions based on divergence**

On the other hand, the $LF_{div}$ and $LF_{LKF}$ suffer from disadvantages.

First, the ability of $LF_{div}$ to detect small leads comes at the price of a higher noise level. Therefore, most of the products use a shape criterion to remove noise. How this could be successfully done on $LF_{div}$, is essentially shown in the $LF_{LKF}$ that do not

contain any noise, but also miss smaller leads. This trade-off between the size of leads and the confidence in them could be adjusted depending on the research question by revisiting the filter of the LKF detection algorithm.

Second, we could only provide uncertainty estimates based on a general tracking error for the accumulated $LF_{accu.\ div}$ even though it is likely that the uncertainty scales with the experienced deformation. The lack of a sufficiently large amount of high-resolution, "ground truth" reference lead data prevented us to study the effects of accumulated uncertainties beyond the

general assumptions we have presented in Section 3.1.3. To overcome this hurdle, future work will involve a comparison with an improved lead dataset based on ICESat-2 (Duncan and Farrell, 2022; Farrell et al., 2020).

Third, to leverage the full potential of the $LF_{LKF}$ as a noise-free, feature-based lead product, they require a more suffocated approach to derive areal lead fractions. Among the two current methods, $LF_{LKF\ pixel}$ estimates aligned more accurately with reference data from $LF_{Heli\_TIR}$ and EM ice thickness observations. However, a future approach for the extraction of divergence

should allow LKFs to have a width of several pixels.

Last, $LF_{div}$ and $LF_{LKF}$ are available every 1-3 days and do not cover the region north of 87°N. This is a restriction inherent to the used Sentinel 1 data and may be overcome by adding other SAR satellites, e.g., the RADARSAT Constellation Mission (Howell et al., 2022). In addition, their time series includes only the recent two decades. Combining SAR datasets from different sensors will reduce those gaps, but, so far, $LF_{PMW}$ and $LF_{MODIS}$ provide more suitable alternatives for climatological

studies reaching back several decades or studies on arctic-wide scales and (sub-)daily time scales.

Taken together, based on our means of comparison, we conclude that $LF_{div}$ and $LF_{LKF}$ combine advantages of several other lead products and are thus a valuable addition to the existing lead products. This corroborates earlier results from Kwok (2002), Kwok and Cunningham (2002), and Kwok (2006) who used RGPS-derived deformation to estimate openings in the ice pack

and divergence-induced new ice formation. Our study could substantially advance the results from Kwok (2002) by using several existing lead products for comparison and by demonstrating that $LF_{div}$ and $LF_{LKF}$ can also reliably indicate the location of leads. In addition, the TIR observations, as well as the open-water fraction from the airborne ice thickness observations (von Albedyll et al., 2021b) provide an independent evaluation for our approach.

**Comparison with previous estimates of lead fraction, lifetime, and width**

Our study also emphasizes the need to distinguish between underlying lead definition and retrieval method when describing lead fractions in the Arctic. Based on $LF_{div}$, $LF_{LKF\ pixel}$, and $LF_{classified\_SAR}$ we provided further evidence for a mean **open-water**





**fraction** in Arctic sea ice in the order of 0.1–1 %. This estimate agrees well with previous results from Kwok (2002) who found a mean open-water fraction of 0.3% based on divergence for the perennial ice cover in the Pacific sector of the Arctic.

In addition, based on the accumulated $LF_{accu. div}$ and and $LF_{Heli, TIR}$, we could also provide supporting arguments for Arctic **lead fractions including thin ice** in the order of 1-3%. Those estimates agree much better with previously reported fractions in the order of a few percent by Wadhams (2000); Reiser et al. (2020).

Furthermore, differences in lead fraction products may also arise from the scale and resolution of the different measurements as small (potentially unresolved) leads dominate (Marsan et al., 2004; Thielke et al., in review).

The observed differences between the lead products of more than 5% are very large compared with the actual physical effects that small increases by, e.g., 1% lead fraction could have on the Arctic climate system (Lüpkes et al., 2008). Therefore, the lead products require careful interpretation with good knowledge of the underlying retrieval methods. Only when considering the specific physical and technical properties of the lead-fraction time series, a confident application is possible.

We briefly showed that lead lifetime and width calculated from $LF_{accu. div}$ fulfill the expected scaling behavior. Our exponential
fit rate of 0.26 1/day to the lead lifetime (3-11 days) is similar to (Hutter et al., 2019, 0.34 1/day for lifetimes > 3 days) who analyzed the RADARSAT Geophysical Processor System (RGPS) dataset. They showed that more than 99% of the LKFs had a lifetime smaller than 12 days which further corroborates our findings. Thus, the error due to our choice of a maximum of 11 accumulation time instances is diminishing.

**Potential application of divergence-based lead fractions**

Derived from SAR data and focusing on the formation of leads, the $LF_{div}$ and $LF_{LKF}$ are well suited to estimate the open-water fraction with high reliability, high spatial resolution and coverage, and moderate temporal coverage. Their ability to identify open water makes the products particularly valuable for applications that deal with all processes happening in leads at the ocean-air interface. Accumulating them to derive $LF_{accu. div}$ opens up an even wider range of applications that also include
leads covered with thin ice. For example, divergence-based lead fractions can be used to study the role of leads in snow loss (Clemens-Sewall et al., 2023), the effect of leads on winter cloud microphysical properties (Saavedra Garfias et al., 2023), or to estimate new ice formation with associated brine release.

The direct and easy way to calculate lead width from $LF_{10x accu. div}$, and their ability to resolve rather small leads, make $LF_{10x accu. div}$ a valuable source of information for the analysis of heat transport through leads. This is because lead width
plays an important role in heat exchange with more efficient heat transfer in rather small leads compared to larger leads (e.g., Andreas and Cash, 1999; Marcq and Weiss, 2012).

The divergence-based lead products can also serve as high-resolution observational reference data for modeling studies that focus on leads and their contribution to the sea ice mass balance (e.g., Ólason et al., 2021; Boutin et al., 2023). Using lead fractions based on divergence could establish a yet missing direct link between changes in drift speeds, deformation rates, and
new ice production.



# 6 Conclusions

Only a small fraction of the Arctic perennial sea ice zone is covered by leads, thin ice, or open water that was created by divergent ice motion. However, those leads are hotspots for many atmospheric, ecological, and oceanic processes in the polar climate system. Precise retrieval techniques are required to observe the small fraction of leads in the sea ice. The aim of

this study was to evaluate SAR-retrieved divergence for estimating lead fractions. Divergence is the driving mechanism of lead formation and we calculated it from sequential SAR images obtained from the Sentinel-1 mission. We derived two lead products from the divergence. The first product, $LF_{div}$, is based on divergence only and identifies leads that formed on the last time step. We accumulated $LF_{div}$ for up to 10 time instances after advecting them to detect also old leads that formed on previous time instances.

The second product, LKF-derived lead fractions ($LF_{LKF}$) are based on LKFs that were identified in the total deformation data using an algorithm by Hutter et al. (2019). This procedure efficiently removes noise and accurately displays the location of newly formed leads.

Evaluating $LF_{div}$ and $LF_{LKF}$ against six other existing lead products, we came to the following conclusions:

(1) Lead fractions based on SAR-derived divergence are valuable additions to the existing lead products as they accurately
capture where and when leads form. Independent of cloud cover, but limited to satellite coverage south of $87°$, they identify open water at high spatial resolution (700 m) and coverage (> 200x200 km), and moderate temporal coverage (1 day).

(2) When accumulated over up to 10 time instances, $LF_{accu.\ div}$ resolve when individual leads formed, were dormant, closed, or re-opened. Combined with a thermodynamic growth model, this allows to reconstruct the lead ice thickness at any
time. This makes $LF_{accu.\ div}$ a valuable tool for estimating the dynamic contribution to the sea ice mass balance.

(3) $LF_{div}$ and $LF_{accu.\ div}$ have plausible mean magnitudes and temporal variability for the open-water fraction and lead fractions including thin ice, respectively. The ability to resolve also small leads with widths as small as 250 m comes at the expense of a higher noise level. In $LF_{LKF}$, noise is efficiently removed, but the area actually covered by leads is reduced, too.

(4) $LF_{div}$ and $LF_{LKF}$ reproduce the temporal variability expected from the large-scale wind forcing, the season, and the consolidation state of the ice pack along the Transpolar Drift. Lead activity is high in the fall and spring and the temporal variability seems to be consistent on scales of 50-150 km around the MOSAiC trajectory, with pronounced differences at smaller scales (10 km).

(5) There are large differences in the lead fractions derived from different products. Any application of them must be un-
dertaken with care and knowledge of the underlying retrieval methods. In addition, other algorithms could be improved based on the comparison with our results.



*Data availability.* Airborne TIR data are available from Pangaea: https://doi.org/10.1594/PANGAEA.951569 (Thielke et al., 2022). Costumer-gridded CryoSat-2, the classified SAR, divergence, and LKF datasets are currently prepared for submission to PANGAEA. Links will be provided during the review process as soon as possible. The standard gridded and trajectory-based AWI CryoSat-2 Sea Ice prod-
uct files (v2.4) are available via ftp://ftp.awi.de/sea_ice/product/cryosat2/v2p4/nh/. This work contains modified Copernicus Sentinel data (2019–2020). Sentinel-1 scenes are available from the Copernicus Open Access Hub (https://scihub.copernicus.eu/dhus/home, last access: 18 August 2021). Code and additional information for the PMW lead fractions are available from https://gitlab.awi.de/public_repository/ mosaic-along-track-amsr2-ice-concentration-and-lead-fraction. Lead fractions from MODIS are available from Pangaea: https://doi.org/10. 1594/PANGAEA.955561 (Willmes et al., 2023).

*Author contributions.* LvA calculated the drift and divergence fields, the divergence-derived lead fractions, and the LKF-derived lead fractions from the provided LKFs. LvA carried out the analysis and prepared the original draft. SH provided the custom-made version of the "Level-3 gridded sea-ice thickness and auxiliary parameters" product. NH developed the directional filter used to remove noise from the divergence fields and detected LKFs in the divergence fields. XT-K updated the algorithm used to derive the PMW lead fractions from AMSR-2 data. LK provided the AMSR-2 lead fractions. DM provided the classified SAR scenes. SW provided the daily MODIS lead products. LT
provided the airborne TIR lead-fraction fields. All authors contributed to the final version of the manuscript.

*Competing interests.* At least one of the (co-)authors is a member of the editorial board of The Cryosphere.

*Acknowledgements.* We thank all MOSAiC participants and people supporting on land for their dedicated work, especially the people involved in the airborne measurement program (Nixdorf et al., 2021). Data used in this manuscript were produced as part of the international MOSAiC project with the tag MOSAiC20192020, Project ID: AWI_PS122_00. This work was mainly funded by the German Federal Ministry of Education and Research (BMBF) through financing the Alfred-Wegener-Institut Helmholtz-Zentrum für Polar- und Meeresforschung (AWI) and the R/V *Polarstern* expedition PS122 under the grant N-2014-H-060_Dethloff and the AWI through its projects: AWI_ICE and AWI_SNOW. LvA and LT acknowledge the support by the Deutsche Forschungsgemeinschaft (DFG) through the International Research Training Group IRTG 1904 ArcTrain (grant 221211316). XT-K acknowledges the support by BMBF through CATS-the changing transpolar system project (03F0776). NH, GS, and CH were supported by the BMBF IceSense (03F0866A & 03F0866B) project and, NH, by a fellow-
ship of the Cooperative Institute for Climate, Ocean, and Ecosystem Studies. MODIS lead detection was funded by the Federal Ministry of Education and Research (BMBF) under grant 03F0831C in the frame of German-Russian cooperation "WTZ RUS: Changing Arctic Transpolar System (CATS)" and by the Deutsche Forschungsgemeinschaft (DFG) in the framework of the priority program "Antarctic Research with comparative investigations in Arctic ice areas" under Grants HE 2740/22 and WI 3314/3. We thank Angela Bliss and Jennifer Hutchings for discussions about sea ice dynamics development throughout MOSAiC.



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
