# Peer review of "Lead fractions from SAR-derived sea ice divergence during MOSAiC"

_The Cryosphere, 2023_

## Referee Comment (RC2)

**Specific comments:**

*Introduction*

L58 – L59: Why is the limitation of daylight not mentioned for visible satellites?

L81: "we aim to present…" I suggest removing the verb "aim" as you successfully present two new lead products in this paper.

*Section 2: "Lead fractions from different retrieval methods"*
*Section 2.1:*

Generally, I like the idea of starting this section with highlighting the different ways how leads can be detected (Section 2.1). It makes a great summary of methods and a good introduction. Nevertheless, there is some disconnect between 2.1 and the satellite products in 2.2 and 2.3. Some subsections contain repetition of information stated (slightly differently) within the physical properties (e.g., L242 – L246 correspond to L118 and following lines). One solution could be mentioning the satellite product using this physical property there and then pointing towards the subchapter or not repeating physical properties and pointing back to the explanation of the physical properties within 2.2 and 2.3?

Why is the concept of LKFs not introduced within (2) Elongated feature?

Figure 1 and Section 2.1 Physical properties of a lead detected by remote sensing are not well connected. Some, but not all titles of the physical properties relate to the Figure. I propose either changing them to make them match or include the numbers from the physical properties into Figure 1.

Other minor comments on Figure 1: The thin ice cover (?) marked with the light blue and crosses is not explained anywhere. On a printed version of the manuscript the yellow and grey writing are hard to read – consider changing colors or making the font bold.

L142: Citation format is not correct.

*Section 2.2:*

L146: Should the title not be plural "Novel lead products based on …"?

L147: I suggest replacing "These" with "Both" for clarity.

L163: Does "area" refer to the radius around the *R/V Polarstern*? If yes, I suggest replacing it.

L171: Does "deformation data" refer to the calculated divergence? Is the directional filter applied to all div data or only div > 0? I assume to all div data (see statement in L181) but maybe that could be made clearer with the naming of the data set.

L181: Why is lead fraction plural in "average lead fractions per grid cell"? Is the result not one lead fraction for the whole grid cell?

L182: I suggest including "rafting" within the results of convergent motion in "indicate closing and ridging".

L190: I personally found it hard to figure out how each step relates to Figure 2. If Figure 2 is here used to illustrate the creation of that data set, maybe include references within the steps of the procedure. If it includes the result, then I would suggest mentioning it after the explanation.

L194: What does "b1" (also in L197) refer to?

L231-234: What exactly is the second quantity? I assume that this paragraph is supposed to give the second quantity as it starts with "Secondly".

L226 – 236: I suggest making it into one paragraph, as "We are computing two different quantities" highlight that both quantities are mentioned within one paragraph. And L235-236 is a solid summary of what the product includes.

*Section 2.3*

L242-246: Repetition of information given in the physical properties of leads (2.1)

*Section 3 Evaluation of lead fraction based on divergence during MOSAiC*
*Section 3.1.1:*

L365: Why are the manually measured widths not included into the Figure 3?

*Section 3.1.2:*

L381-382: According to Figure 4b) the lifetime always follows an exponential fit (with all lifetimes and with 3-11 days lifetime)? Why does the text only mention the latter? Also, why observing the lifetime after 3 days only if 33% of the leads are present for two days? (L383)? I am not sure if displaying both fits in the Figure 4b) is necessary and if it is necessary, this information is lost on me.

L385: How many leads are 2% of the leads? All mentioned numbers are given in relative numbers or percentages, and I suggest mentioning somewhere how many leads occur in your analysis.

L393-394: Why is the exponent of the linear fit not mentioned? The exponent is not mentioned here, nor is it discussed in comparison to other exponents derived for the Arctic sea ice based on remote sensing (e.g., Wernecke and Kaleschke (2015), Marcq and Weiss (2012)) later in the article.

L400-401: I disagree with the correlation between the lead width variability and the number of lead pixels as a Pearson R value from 0.26 is a low R value. The correlation might be strong in October and November, but there is no effect between April and May, which is a time of strong variability. Additionally, periods of similar variability as October and November do not show an increase in number of lead pixels (e.g., December).

*Section 3.1.3:*

L414: Are "time instances" the same as "time steps" (used in L390)? From here on onwards I only recognized the usage of "time instance" and was sometimes confused what the difference between a "time instance" and a "time step" would be.

L418: Does "no geolocation errors" mean that there are no errors due to advection?

L421: I assume that the "tracking uncertainty" is the before mentioned "tracking error"?

L422: It was hard for me to identify the unit of the lead fraction uncertainty. Maybe it would be clearer if the unit would be written directly after the value as a unit and not in words. Similar to L423 where it would be easier with 56-122 m day$^{-1}$.

L431: What is the "threshold for lead-ice thickness set by the research question"?

After the end of this section, it is still hard for me to understand the uncertainty of your products, e.g., what does this mean for people using this product and how this uncertainty compares to other available products.

*Section 3.1.4:*

Figure 6: Panel a): It took me a significant amount of time to identify the "no accumulation" and "10x accumulation" line and understand that this panel highlights how the accumulation effects the lead activity. Maybe these lines need color, or the gray lines (1x-9x) in-between are not needed.

*Section 3.1.5:*

Figure 6: Panel b) I assume the goal of this panel is to show how the area influences the lead fraction. For consistency I would remove the gray 0-10x accumulated lines, because all other lines display 5x accumulated. Additionally, the dark purple and black line are basically not distinguishable (neither in print-out nor on screen), which is unfortunate as this is the line especially mentioned in the text.

L474-475: I suggest including a reference for localized and intermittent nature of deformation. Additionally, I was wondering if the mentioned effect within the smallest radius results from the *R/V Polarstern* influencing the local ice.

*Section 4: Comparison of different lead products*
*Section title of 4.2 "Temporal variability of different lead products"*

This section has several inconsistencies within its organization:
- The title of the section (L529) is not properly represented in the topic sentence (L530-531) of the section with "(1) the temporal variability and (2) the temporal resolution and coverage".
- The name of (1) within the topic sentence (L530) is inconsistent with the later used title for part (1) (L533)
- The name of (2) within the topic sentence (L530) is also inconsistent with the later used title for part (2) (L575)

This makes it hard for the reader to figure out what exactly is discussed in this section and where it is discussed. I also wonder why here the numeration of (1) and (2) is used, while other sections have a second-level subsection (e.g., 3.1 with) subsections ranging from 3.1.1 to 3.15.

Figure 7: I suggest calculating the ice concentration from 0 to 1 instead of the display with "1-ice concentration" as this feels unintuitive. Additionally, the text only mentions the ice concentration and quick readers might miss the small gray label.

L544-545: For me it is not obvious why the last sentence of this paragraph ("In fact, ….") explains the difference between the classified SAR and the divergence products highlighted in blue in Figure 7. The $LF_{PMW}$ also features a blue area – how does that fit into the picture?

Figure 7: Does it make sense to connect the dots with a dashed line for $LF_{Heli\_TIR}$? Why are a few C2 absolute dots not connected?

*Section 4.3*

L587 and L614: Why do these headings not result in their own subsection 4.3.1 and 4.3.2 respectively?

Figure 8: Is there a way to highlight the four lead pixel within the MODIS subplot (f)? On print-out they are not visible at all and on-screen only with zooming in a lot. Otherwise, a comment in the Figure description could help as well.

Figure 9: The comparison between e) and f), for example, is difficult visually as red is a more vibrant color compared to the muted green for the MODIS subplot. I suggest making the colors for leads the same or at least perceptually similar. The same comment applies to g) and h). Why does the colormap change?

L587-590: Where do the measurements from the leads come from? What is their uncertainty?

L643-645: I do not understand why missing leads is a reason for a comparably high lead fraction of $LF_{CS2}$?

L646-648: A paragraph needs at least two sentences. Additionally, there is no summary of the performance from both case studies.

*Section 5: Discussion*

L652-653: The sentence introducing the advantages should go into the section where you address these advantages. Otherwise, the impression could arise that these four advantages could relate to the four bold titles throughout this section.

L656, L686 L715, L735: Why are these no subsections but rather bold titles?

L659-660: I assume this sentence is about the large-scale ice strength as we miss detailed information of the ice on centimeter to meter scale, which are scales mainly associated with mechanical properties of sea ice. Maybe phrasing it more precise would be helpful.

L687: One sentence is not a full paragraph. Additionally, "on the other hand" generally needs first a mention of "on the one hand".

L688: I suggest adding an "other" in front of "products" as I assume that the usage of a shape criterion is compared to $LF_{div}$.

L692-694: It would be nice if there would also be a comment on the uncertainty. After the uncertainty section and this paragraph in the discussion it is still unclear to me how you rate your uncertainty and how strong it is (compared to other products).

L697: I assume "suffocated" is the wrong word here.

L730: This result of the exponent is not mentioned in the results section.

According to the bold title in L715 this section should include a discussion of the lead width. This discussion does not happen apart from stating that it follows an exponential fit. As there are several studies about lead-width distribution cited in this paper I suggest that you discuss your result of the lead-width distribution quickly with other research. Otherwise, there seems to be no assessment of the general lead width detected by your methods.

*Section 6: Conclusion*

L757 and L760: The end of the paragraph is odd as "the first product" (L757) and "the second product" (L760) should be in one paragraph. One suggestion would be to make the paragraph break after "Sentinel-1 mission" and combine all following lines until the enumeration (L764) into one paragraph.

L758 and L768: What is the difference between a "time step" and "time instance"? I know I asked that above as well, but especially in the conclusions it needs to be clear if this is the same or something different as readers might only read the conclusions.

**Technical comments:**

The use of compound adjectives varies within the paper and leads to inconsistent use:
   a) Inconsistencies within the same word group; examples:
      - sea ice cover (e.g., L19), sea ice dynamics (e.g., L24) and similar are mainly written without turning "sea ice" into a compound adjective ("sea-ice"), but sometimes "sea-ice" is used as a compound adjective (e.g., L82 sea-ice divergence)
      - "Lead-fraction retrievals" (L66, L91) versus "lead fraction dataset" (L73)
      - "EM ice-thickness measurements" (L515) versus "airborne ice thickness measurements" (L514-515)
   b) Inconsistencies across the article
      Compound adjectives are for example used for ice-covered leads (e.g., L28)

I know that some people assume compound adjectives to be a matter of personal style.

Some numbers written in the text do not follow the TC style guide (https://www.the-cryosphere.net/submission.html#math). One example is "6 existing lead products" (L498).

Not all (e.g., …) references in the manuscript include the necessary comma after "e.g." (e.g., L43, L45 and L143)

L142: Citation format is not correct.

L226: LKLKF is not formatted in the right way.

L336: Citation format is not correct.

L371: There is a space missing between the word "studies" and the start of the citation.

L393-394: There is a comma between the number and the unit in three cases.

L419: L should be replaced with $\Delta$L.

Figure 4 and Figure 5: The caption should probably be "mean ($\pm$ standard deviation)

L483: LFdiv should be LF$_{div}$.

L489: I assume "as" is supposed to be "has".

L512: 0.05 misses the "%" symbol.

Table 1 – first row: Coefficient of variation has only one number after the decimal, should have two for consistency with other rows.

L607: I suggest exchanging the verb "leads" to another verb as this is slightly confusing for the reader.

L634: There is the word "Figure" missing in the brackets.

L668: "on" should be replaced with "in".

---

## Author Comment (AC1)

Response to Reviewer 1

Thank you very much for reviewing our manuscript and raising the interesting idea of combining the two SAR-based approaches (i.e., from SAR-based ice motion and from the SAR image classification) into an improved SAR-based lead fraction product. In this manuscript, we focus on the different approaches to find out their advantages and disadvantages. A combined approach would potentially bring both the advantages and disadvantages together, so a merged approach requires careful evaluation of the present results and could be a subject of a future study. We suggest to add the following comment to the discussion in the manuscript:

"The comparison of lead fraction products presented here allows us to explore ways to mitigate their drawbacks by combining them. A promising approach could be to merge the two SAR-based methods, LF $_{accu.\ div}$ and LF$_{SAR}$, within a single algorithm, as they both are based on the same data source. Leveraging the higher resolution of LF$_{SAR}$ (80 m compared to 700 m), we can use LF$_{SAR}$ to precisely pinpoint the location of leads when LF $_{accu.\ div}$ indicates their presence. Simultaneously, LF $_{accu.\ div}$ and LF$_{SAR}$ can be used as pre-filter for the respective other, replacing or relaxing the existing, potentially stricter filters. This combined approach has the potential to reduce the number of misclassifications and to suppress noise but may also bring the disadvantages of both methods together, so a merged approach requires careful evaluation of the present results and could be a subject of further studies."

Our answers to your minor comments (black) are given below in blue.

Minor comments:
Line 23. I suggest replacing "fast new ice formation" with "rapid new ice formation" to avoid any confusion with "fast ice" term.
Good point. We replaced it.

Line 161. Please explain why HV channel was not used for ice motion detection.
We use an existing sea ice drift algorithm that was, so far, mainly tested and applied to co-polarized SAR (e.g., Hollands 2012, Griebel and Dierking, 2017, including previous work of the author, e.g., von Albedyll et al. 2021). For consistency with previous estimates of uncertainty, we use the HH channel. HH has the advantage that the magnitude of co-polarization is larger than of cross-polarization which is notably affected with thermal, scalloping, and speckle noise. However, we are aware that there are indications that cross-polarized images record ice structures more clearly provided that the noise level is sufficiently low (Komarov and Barber, 2014). Nevertheless, due to the lower signal to noise ratio in the HV, pattern matching in HV can be prone to errors and would require additional extensive quality checks before being ready to use.

Hollands, T. and Dierking, W.: Performance of a multiscale correlation algorithm for the estimation of sea-ice drift from SAR images: initial results, Annals of Glaciology, 52, 311–317, https://doi.org/10.3189/172756411795931462, 2011.
Griebel, J. and Dierking, W.: Impact of Sea Ice Drift Retrieval Errors, Discretization and Grid Type on Calculations of Ice Deformation, Remote Sensing, 10, 393, https://doi.org/10.3390/rs10030393, 2018.
von Albedyll, L., Haas, C., and Dierking, W.: Linking sea ice deformation to ice thickness redistribution using high-resolution satellite and airborne observations, The Cryosphere, 15, 2167–2186, https://doi.org/10.5194/tc-15-2167-2021, 2021.
Komarov A. S., and Barber D. G. "Sea ice motion tracking from sequential dual-pol RADARSAT-2 images." IEEE Trans. Geosci. Remote Sens. 2014, vol. 52, no. 1, pp. 121–136. doi:10.1109/TGRS.2012.2236845

Line 194. Beginning of the sentence with "b1 each lead-fraction …" does not seem to be correct.
Yes, b1 should be replaced with "We advect"

Line 197. "Next, b1 the lead" does not sound correct
Here, as well, b1 should be replaced with "we advect"

Line 226. "LFLKF" -> "$LF_{LKF}$".
Done

Line 393-394. Remove comma in "56-112,m", "56,m", "1500,m".
Done

Equations 2-3. It seems that lead fraction uncertainty (sigma_LF) should be dimensionless. However, from equations 2-3 it seems to have a unit [$s^{-1}$]. Please explain.
Thanks for spotting this inconsistency. We suggest to clarify in the text:

To quantify the uncertainty of the lead fraction magnitude, we first simplify the calculation of the dimensionless lead fractions by omitting the time step information. The lead fractions can also be expressed as the ratio of the difference in displacement (Δ Disp) and the grid cell length scale(L=700 m), thus: LF = Δ Disp/L . Based on the simplified equation, we calculate the uncertainty of the lead fraction magnitude of a single time step from the tracking uncertainty using error propagation assuming no geolocation errors following Dierking et al. (2020). Adapting their equation 17, the uncertainty of the lead fractions σLF is given by the ratio of the tracking uncertainty σtr and spatial L scale: σLF =√2$\sigma_{tr}$/L. With a tracking uncertainty of $\sigma_{tr}$ = 40 – 80 m (Hollands and Dierking, 2011) and a spatial scale of L=700 m this results in an uncertainty of $\sigma_{LF}$ = 0.08-0.16 for a single lead fraction pixel. Translated into lead width, this corresponds to 56–112 m per day when assuming that the lead has opened only along one dimension. For the accumulated lead fractions, we add up the absolute errors of each time step assuming that they are independent from each other. Averaging over larger spatial scales assuming independent errors, we quantify the uncertainty of the $LF_{accu.\ div}$ with the standard error of the mean lead fractions:
$\sigma_{LF\ k\ accu.\ div}$ = k · $\sigma_{LF}$/(√n)
where k is the number of accumulations and n is the number of pixels that fit into circles with radius 10 km, 50 km, 100 km, and 150 km. For $LF_{5x\ accu.\ div}$, this calculation yields uncertainties for the spatially averaged lead fractions of $\sigma_{LF\ 5x\ accu.\ div}$ = 0.019–0.038 (10 km), $\sigma_{LF\ 5x\ accu.\ div}$ = 0.004−0.008 (50 km), $\sigma_{LF\ 5x\ accu.\ div}$ = 0.002−0.004 (100 km) and $\sigma_{LF\ 5x\ accu.\ div}$ = 0.001−0.003 (150 km).

Equation 3. It is not clear why parentheses in "(n)" are required.
We removed the parentheses.

Line 421. "spatial scale of ΔL=700 m" -> "spatial scale of L=700 m"
We removed Δ as it is not needed.

Fig. 5a. It is very difficult to distinguish vertical blue bars and the blue line as they are both blue.
We have revised this part of the manuscript and removed Figure 5a completely.

Line 512. "by two magnitudes" -> "by two orders of magnitude".
Done

Lines 560-561. "LFPMW"->"LF$_{PMW}$".
Done

---

## Author Comment (AC2)

**Reviewer 2**

Thank you very much for all the time you spent evaluating our manuscript. We appreciate your comments and think they can really improve the consistency and clarity of the paper. Please find our responses to your specific comments (black) below in blue.

**General comments:**

My main points of the review of the manuscript are about the "presentation quality". In terms of the presentation quality, I am mainly concerned about inconsistent use of terminology within the paper and inconsistent structures of subsections within sections as well as (sub-) section names not accurately reflecting the content of the sections. Overall, this could be a minor issue but with a manuscript of this length it is easy to get lost. Examples of both are in the specific comments organized by section and technical comments, which are in the attached pdf. I want to highlight that while there are several comments, I do not consider any of these as a major scientific problem. I hope the comments help the authors to review the manuscript and enhance the clarity of it.

We followed your specific and technical comments to improve the structure of the manuscript. In addition to many small changes, the main changes of the structure are:

- Better links between the subsections of "2.1 physical properties" and the products described in 2.2. and 2.3. and Figure 1.
- More subsections in Section 4 (Comparison) and Section 5 (Discussion) instead of bold headlines.
- More consistency between headlines, topic sentences, and content of the sections.

**Specific comments:**

Introduction
L58 – L59: Why is the limitation of daylight not mentioned for visible satellites?
We added the point. "low temporal coverage due to clouds **or the absence of daylight**"

L81: "we aim to present…" I suggest removing the verb "aim" as you successfully
Changed, also for following sentence: **First, we present** and evaluate … **Second, we present** and analyze a …

Section 2: "Lead fractions from different retrieval methods"
Section 2.1:
Generally, I like the idea of starting this section with highlighting the different ways how leads can be detected (Section 2.1). It makes a great summary of methods and a good introduction. Nevertheless, there is some disconnect between 2.1 and the satellite products in 2.2 and 2.3. Some subsections contain repetition of information stated (slightly differently) within the physical properties (e.g., L242– L246 correspond to L118 and following lines). One solution could be mentioning the satellite product using this physical property there and then pointing towards the subchapter or not repeating physical properties and pointing back to the explanation of the physical properties within 2.2 and 2.3?
We added references and links in the "physical properties" sections to the "satellite products" sections and vice versa to make the connection much more obvious. We reduced the description of physical properties in the product descriptions to the bare minimum and refer to the "physical properties" sections instead.

Why is the concept of LKFs not introduced within (2) Elongated feature?
We had overlooked this and changed it in the text.

Figure 1 and Section 2.1 Physical properties of a lead detected by remote sensing are not well connected. Some, but not all titles of the physical properties relate to the Figure. I propose either changing them to make them match or include the numbers from the physical properties into Figure Other minor comments on Figure 1: The thin ice cover (?) marked with the light blue and crosses is not explained anywhere. On a printed version of the manuscript the yellow and grey writting are hard to read – consider changing colors or making the font bold.
We have followed your suggestions and improved Figure 1 accordingly.
  - Text fond is bold
  - Numbers refer to subsection in which property is discussed
  - Names in figure are (almost) identical to the headlines of the subsections
  - Thin ice cover is explained in the caption.

L142: Citation format is not correct.
Changed

Section 2.2:
L146: Should the title not be plural "Novel lead products based on …"?
Yes -> changed

L147: I suggest replacing "These" with "Both" for clarity.
Done

L163: Does "area" refer to the radius around the R/V Polarstern? If yes, I suggest replacing it.
We clarified to: "The scenes were acquired along the drift track of the MOSAiC expedition, are centered around R/V Polarstern, and have typically a side length of 200-300 km."

L171: Does "deformation data" refer to the calculated divergence? Is the directional filter applied to all div data or only div > 0? I assume to all div data (see statement in L181) but maybe that could be made clearer with the naming of the data set.
We have clarified this by moving the text about the separation in divergence and convergence to a point in the text when all calculations on the "full" divergence dataset are already described. Please see Section 2.2.1 for the new version.

L181: Why is lead fraction plural in "average lead fractions per grid cell"? Is the result not one lead fraction for the whole grid cell?
Changed it to singular

L182: I suggest including "rafting" within the results of convergent motion in "indicate closing and ridging".
Added "rafting" as suggested.

L190: I personally found it hard to figure out how each step relates to Figure 2. If Figure 2 is here used to illustrate the creation of that data set, maybe include references within the steps of the procedure. If it includes the result, then I would suggest mentioning it after the explanation.
We agree that the first mention of the Figure was too early and moved it downwards with some additional explanations: "Next, we advect the lead fractions originally based on SAR scenes from March 15/16 to its location on March 16, March 17, ..., and March 28. **Figure 2 displays the lead**

**fractions from March 20/23 to March 26/27 that were all advected to March 27, 2020.** We save all advected lead …

L194: What does "b1" (also in L197) refer to?
We replaced it with "we advect"

L231-234: What exactly is the second quantity? I assume that this paragraph is supposed to give the second quantity as it starts with "Secondly".
L226 – 236: I suggest making it into one paragraph, as "We are computing two different quantities" highlight that both quantities are mentioned within one paragraph. And L235-236 is a solid summary of what the product includes.
We follow your suggestion and combine all into one paragraph. In addition, we added a sentence before the paragraph, naming both quantities. Also, we have formatted the text to make clear that there are two quantities. The paragraph reads now:

We compute two different quantities from the LKF dataset: (1) $LF_{LKF}$ *lead fractions* and (2) $LF_{LKF}$ *binary lead pixel numbers*. For the $LF_{LKF}$ *lead fractions*, […]. For the $LF_{LKF}$ *binary lead pixel numbers*, a pixel is […]

Section 2.3
L242-246: Repetition of information given in the physical properties of leads (2.1)
We agree and have merged the text with the physical properties of leads description. We have removed this part in this section and refer to the Physical Property Section instead.

Section 3 Evaluation of lead fraction based on divergence during MOSAiC
Section 3.1.1:
L365: Why are the manually measured widths not included into the Figure 3?
That's a good suggestion. We have added the manually measured widths into the figure.

Section 3.1.2:
L381-382: According to Figure 4b) the lifetime always follows an exponential fit (with all lifetimes and with 3-11 days lifetime)? Why does the text only mention the latter? Also, why observing the lifetime after 3 days only if 33% of the leads are present for two days? (L383)? I am not sure if displaying both fits in the Figure 4b) is necessary and if it is necessary, this information is lost on me.
L385: How many leads are 2% of the leads? All mentioned numbers are given in relative numbers or percentages, and I suggest mentioning somewhere how many leads occur in your analysis.
L393-394: Why is the exponent of the linear fit not mentioned? The exponent is not mentioned here, nor is it discussed in comparison to other exponents derived for the Arctic sea ice based on remote sensing (e.g., Wernecke and Kaleschke (2015), Marcq and Weiss (2012)) later in the article.
L400-401: I disagree with the correlation between the lead width variability and the number of lead pixels as a Pearson R value from 0.26 is a low R value. The correlation might be strong in October and November, but there is no effect between April and May, which is a time of strong variability. Additionally, periods of similar variability as October and November do not show an increase in number of lead pixels (e.g., December).
Guided by your questions, we have revisited the analysis of the statistics section and improved it. In this context, we have also changed the way how we count the leads avoiding any double counting. Therefore, plots, numbers and text have slightly changed. As suggested by you, we provide now the total number of analyzed leads. We show the fit of all lifetimes and discuss the slope in the results and discussion section. We have also revisited the lead widths and changed the original plot to one in loglog space. We discuss now the lead width exponent in the discussion section in the context of other studies. Further, we have decided to remove the temporal variability of the lead widths

because we agree with you that more detailed research is necessary to understand the relationship between lead width and lead occurrence.

Section 3.1.3:
L414: Are "time instances" the same as "time steps" (used in L390)? From here on onwards I only recognized the usage of "time instance" and was sometimes confused what the difference between a "time instance" and a "time step" would be.
Thanks for pointing out this inconsistency. We refer to a "time instance" when we mean a pair of two sequential SAR images, e.g., March 26/27. We clarified, when needed, in the text for all remaining "time steps" whether we meant "time instance" or "time difference".

L418: Does "no geolocation errors" mean that there are no errors due to advection?
The geolocation error describes an error associated with the input SAR images. It is the error in the position of reference points, i.e., the vertices of the SAR image. Such errors in the position in a SAR image can be caused by the inaccuracies of the parameters describing the satellite orbit as a function of space and time and they are usually uniform across the image (Dierking et al. 2020). The interested reader may find all details in the indicated publication.

Dierking, W., Stern, H. L., and Hutchings, J. K.: Estimating statistical errors in retrievals of ice velocity and deformation parameters from satellite images and buoy arrays, The Cryosphere, 14, 2999–3016, https://doi.org/10.5194/tc-14-2999-2020, 2020.

L421: I assume that the "tracking uncertainty" is the before mentioned "tracking error"?
Yes. We replaced all "tracking error" with "tracking uncertainty"

L422: It was hard for me to identify the unit of the lead fraction uncertainty. Maybe it would be clearer if the unit would be written directly after the value as a unit and not in words. Similar to L423 where it would be easier with 56-122 m day-1.
Thanks for your suggestion. We have revised the uncertainty calculations and made sure that the uncertainty is given as a dimensionless quantity like the lead fractions.

L431: What is the "threshold for lead-ice thickness set by the research question"?
With this statement we wanted to emphasize that different studies (with different research questions) may define different thresholds up to which thickness a lead is still considered as lead. However, we understand that our original sentence may have been unclear and we replaced it with: "In winter, thermodynamic growth sets an upper limit on the accumulation time instances because a lead is "thermodynamically closed" after several days of ice growth."

After the end of this section, it is still hard for me to understand the uncertainty of your products, e.g., what does this mean for people using this product and how this uncertainty compares to other available products.

Thanks for mentioning the lack of a more detailed uncertainty discussion. We have now added a new section (5.1 Uncertainties of the lead fraction products) to discuss our uncertainty estimates and give absolute and relative uncertainties for the time series mean fraction. We also added a short paragraph about comparing the uncertainties of the different products but emphasize that this is highly complex due to the different nature of the products. Our comparison forms a basis for improving and reassessing the uncertainty estimates of all products.

Section 3.1.4:

Figure 6: Panel a): It took me a significant amount of time to identify the "no accumulation" and "10x accumulation" line and understand that this panel highlights how the accumulation effects the lead activity. Maybe these lines need color, or the gray lines (1x-9x) in-between are not needed.
We added color to the lines to ease the interpretation of the graph.

Section 3.1.5:
Figure 6: Panel b) I assume the goal of this panel is to show how the area influences the lead fraction.
For consistency I would remove the gray 0-10x accumulated lines, because all other lines display 5x accumulated. Additionally, the dark purple and black line are basically not distinguishable (neither in print-out nor on screen), which is unfortunate as this is the line especially mentioned in the text.
We have removed the thin gray lines and we have changed the line colors.

L474-475: I suggest including a reference for localized and intermittent nature of deformation.
We have added two references and rephrased: "On the smaller scale, the localized and intermittent nature of deformation (Marsan et al. 2004, Hutchings et al. 2011) starts to become apparent with localized lead events hitting (or missing) the smaller area."

Marsan, D., Stern, H., Lindsay, R., and Weiss, J.: Scale Dependence and Localization of the Deformation of Arctic Sea Ice, Physical Review Letters, 93, https://doi.org/10.1103/physrevlett.93.178501, 2004.

Hutchings, J. K., Roberts, A., Geiger, C. A., and Richter-Menge, J.: Spatial and temporal characterization of sea-ice deformation, Annals of Glaciology, 52, 360–368, https://doi.org/10.3189/172756411795931769, 2011

Additionally, I was wondering if the mentioned effect within the smallest radius results from the R/V Polarstern influencing the local ice.
Thanks for raising this interesting question. Most likely, *Polarstern* had a very local impact on the sea ice dynamics as it acts like an additional mass on the mechanical system. Due its height larger than any pressure ridge, it locally absorbs more kinetic energy to the ice-ocean system. If this had significant impact, we would expect a difference in the local drift the ice in the wider vicinity. We could not see any influence of *Polarstern* when comparing deformation in a 5 km radius around *Polarstern* with deformation in other 5 km radius circles within 50 km distance (Krumpen et al. 2021, Figure 3 and 17). We thus conclude that the influence of the ship is, if present, negligible on scales larger than 5 km.

Krumpen, T., von Albedyll, L., Goessling, H. F., Hendricks, S., Juhls, B., Spreen, G., Willmes, S., Belter, H. J., Dethloff, K., Haas, C., Kaleschke, L., Katlein, C., Tian-Kunze, X., Ricker, R., Rostosky, P., Rückert, J., Singha, S., and Sokolova, J.: MOSAiC drift expedition from October 2019 to July 2020: sea ice conditions from space and comparison with previous years, The Cryosphere, 15, 3897–3920, https://doi.org/10.5194/tc-15-3897-2021, 2021b.

Section 4: Comparison of different lead products
Section title of 4.2 "Temporal variability of different lead products"
This section has several inconsistencies within its organization:
- The title of the section (L529) is not properly represented in the topic sentence (L530-531) of the section with "(1) the temporal variability and (2) the temporal resolution and coverage".
- The name of (1) within the topic sentence (L530) is inconsistent with the later used title for part (1) (L533)
- The name of (2) within the topic sentence (L530) is also inconsistent with the later used title

for part (2) (L575)

This makes it hard for the reader to figure out what exactly is discussed in this section and where it is discussed. I also wonder why here the numeration of (1) and (2) is used, while other sections have a second-level subsection (e.g., 3.1 with) subsections ranging from 3.1.1 to 3.15.

Thanks for pointing this out. We have changed "Temporal and spatial coverage of different lead products" into an own subsection and changed the topic sentences of Section 4 and Section 4.2 respectively.

Figure 7: I suggest calculating the ice concentration from 0 to 1 instead of the display with "1-ice concentration" as this feels unintuitive. Additionally, the text only mentions the ice concentration and quick readers might miss the small gray label.

We aim for leads, indicated by decreases in ice concentration, to appear as peaks, aligning them with the patterns observed in other time series. To ensure clarity, we have re-labeled the y-axis as 'open-water fraction' and explained in the caption that this fraction is derived from 1-ice concentration.

Figure 7: Does it make sense to connect the dots with a dashed line for LFHeli_TIR? Why are a few C2 absolute dots not connected?

We agree that it does not make sense and have removed the dashed lines for $LF_{Heli\_TIR}$. Points are not connected if there is a NaN in between, i.e., the temporal resolution could be higher but there is a data gap.

Section 4.3

L587 and L614: Why do these headings not result in their own subsection 4.3.1 and 4.3.2 respectively?

Good suggestion that we are happy to follow. We have created subsections for them.

Figure 8: Is there a way to highlight the four lead pixel within the MODIS subplot (f)? On print-out they are not visible at all and on-screen only with zooming in a lot. Otherwise, a comment in the Figure description could help as well.

Figure 9: The comparison between e) and f), for example, is difficult visually as red is a more vibrant color compared to the muted green for the MODIS subplot. I suggest making the colors for leads the same or at least perceptually similar. The same comment applies to g) and h). Why does the colormap change?

Thank you for your feedback on improving our figures. We now consistently use red to highlight leads in all our plots. The variety in colormaps is due to the differing value ranges across various products. Our focus here is on the precision with which the products pinpoint the location of the leads, rather than the magnitude of the lead fraction. Therefore, we've opted to maintain different colormaps. We believe the leads in the MODIS subplot will now be more prominent with the use of red. Additionally, we've included two explanatory sentences in the figure caption for clarity: "The $LF_{MODIS}$ in panel (f) are strongly affected by clouds (black). The few lead pixels (red) are located close to the center of the circle."

L587-590: Where do the measurements from the leads come from? What is their uncertainty?

All quoted numbers are manually measured on the SAR image with a resolution of 50 m. The uncertainty is 1 pixel, i.e., 50 m. We have added this to the text: "(manually measured on SAR image with 50 m resolution)"

L643-645: I do not understand why missing leads is a reason for a comparably high lead fraction of LFCS2?

Our statement was probably not clear here. The $LF_{CS2}$ of March 27 are not high compared to the whole $LF_{CS2}$ time series; they are not a clear peak, even though there were a lot of leads present compared to other days of the time series. For example, the $LF_{CS2}$ of Nov 2 is higher/in the same order of magnitude even though there was only two leads present. We have rephrased this part:

The $LF_{CS2}$ indicate a lead fraction of 8.4% which is lower than for November 2, 2019, despite more leads being present. Even though the swaths indicate some coverage of the leads, the valid waveforms for the lead pixels are rather low. We speculate that the very small leads might have been missed since the corresponding waveforms are also influenced by surrounding sea ice, and are subsequently classified as mixed surface type and intentionally removed from the processing. The sufficient lead area fraction within the radar footprint needed for a lead waveform classification is not known and likely depends on the actual geometry and specific lead radar backscatter characteristics. Lead waveform classification however is possible in the presence of sea ice, thus it is reasonable to assume that the rate of lead detections of radar altimeter data still overestimates the true lead area fraction.

L646-648: A paragraph needs at least two sentences. Additionally, there is no summary of the performance from both case studies.
We extended the paragraph with a summary sentence:

"We conclude that a spatial analysis of lead products, e.g., a visual comparison with higher-resolution optical or SAR data and a plausibility check of the shape and stability of detected leads provides relevant information about the ability of products to indicate leads on the chosen temporal and spatial resolution. The SAR-based lead fractions ($LF_{div}$, $LF_{LKF}$, $LF_{5x\ accu.\ div}$, $LF_{classified\ SAR}$) perform best in locating the leads while the other sensors suffer from low coverage due to clouds or no overpasses, the presence of thin surrounding ice or too small leads."

Section 5: Discussion
L652-653: The sentence introducing the advantages should go into the section where you address these advantages. Otherwise, the impression could arise that these four advantages could relate to the four bold titles throughout this section.
L656, L686 L715, L735: Why are these no subsections but rather bold titles?
We follow your suggestions and have changed the structure of the discussion into:
   - Subsections and subsubsections
   - Extended the introduction sentence of the discussion
   - Moved the sentence with the advantages and disadvantages to the respective sections

Here is the structure as described in the beginning of the discussion:

The objective of this study was to analyze lead products based on divergence. We have calculated two lead products from the divergence: divergence-derived (accumulated) lead fractions ($LF_{div}$) and LKF-derived lead fractions ($LF_{LKF}$). In the following, we discuss the advantages and disadvantages of the LFdiv and LFLKF compared to the other lead datasets (Section 5.2), compare our lead statistics with those of other studies (Section 5.3), and conclude with presenting potential applications of lead fractions based on divergence (Section 5.4).

L659-660: I assume this sentence is about the large-scale ice strength as we miss detailed information of the ice on centimeter to meter scale, which are scales mainly associated with mechanical properties of sea ice. Maybe phrasing it more precise would be helpful.

Thanks! Yes, indeed we had large-scale mechanical behavior in mind. We rephrase to:

"The knowledge of the deformation history of a lead enriches our understanding of the large-scale ice strength and preferred means of sea ice redistribution."

L687: One sentence is not a full paragraph. Addtionally, "on the other hand" generally needs first a mention of "on the one hand".
We removed the sentences as the heading gives enough orientation about the content of the section.

L688: I suggest adding an "other" in front of "products" as I assume that the usage of a shape criterion is compared to $LF_{div}$.
Done

L692-694: It would be nice if there would also be a comment on the uncertainty. After the uncertainty section and this paragraph in the discussion it is still unclear to me how you rate your uncertainty and how strong it is (compared to other products).
We have added a full section on uncertainties to the discussion that addresses how the uncertainty is rated against other products (text see above to your other question about uncertainty).

L697: I assume "suffocated" is the wrong word here.
Changed to "sophisticated"

L730: This result of the exponent is not mentioned in the results section.
We added this information to the results:
"The lifetime distribution follows a negative exponential fit (Figure 4b, left y-axis) with an exponent of 0.39 day$^{-1}$."

According to the bold title in L715 this section should include a discussion of the lead width. This discussion does not happen apart from stating that it follows an exponential fit. As there are several studies about lead-width distribution cited in this paper I suggest that you discuss your result of the lead-width distribution quickly with other research. Otherwise, there seems to be no assessment of the general lead width detected by your methods.
Thanks for making us aware of this gap. We have added now a comparison:
"For the lead width scaling, we determined a power-law exponent of 2.55 across a range of 50 to 1200 m by calculating a linear fit in a log-logplot. This exponent is at the higher end of the 1.4 to 2.6 range reported in the literature, as detailed in Muchow et al. 2021 (their Table 3).

Muchow, M., Schmitt, A. U., and Kaleschke, L.: A lead-width distribution for Antarctic sea ice: a case study for the Weddell Sea with high-resolution Sentinel-2 images, The Cryosphere, 15, 4527–4537, https://doi.org/10.5194/tc-15-4527-2021, 2021.

Section 6: Conclusion
L757 and L760: The end of the paragraph is odd as "the first product" (L757) and "the second product" (L760) should be in one paragraph. One suggestion would be to make the paragraph break after "Sentinel-1 mission" and combine all following lines until the enumeration (L764) into one paragraph.
We followed your suggestion.

L758 and L768: What is the difference between a "time step" and "time instance"? I know I asked that above as well, but especially in the conclusions it needs to be clear if this is the same or something different as readers might only read the conclusions.
We have changed "time step" to "time instance" for consistency.

**Technical comments:**

The use of compound adjectives varies within the paper and leads to inconsistent use:
a) Inconsistencies within the same word group; examples:
- sea ice cover (e.g., L19), sea ice dynamics (e.g., L24) and similar are mainly written without turning "sea ice" into a compound adjective ("sea-ice"), but sometimes "sea-ice" is used as a compound adjective (e.g., L82 sea-ice divergence)
- "Lead-fraction retrievals" (L66, L91) versus "lead fraction dataset" (L73)
- "EM ice-thickness measurements" (L515) versus "airborne ice thickness measurements" (L514-515)
b) Inconsistencies across the article
Compound adjectives are for example used for ice-covered leads (e.g., L28)
I know that some people assume compound adjectives to be a matter of personal style.
Thank you for bringing the inconsistencies to our attention. We have chosen to use "sea ice," "lead fraction," "ice thickness," "open water," "synthetic aperture radar," and "ice covered leads" without hyphens because these terms are well-understood and thus partly standard in our field. For product names such as "Level-3 gridded sea-ice thickness and auxiliary parameters," we will maintain their original format. For less well-known terms like "divergence-derived," "SAR-derived," "divergence-based," "1-d kernel," "Classified-SAR," "pixel-based," "divergence-induced," and so on, we use a hyphen. We're open to aligning with the journal's style guidelines on this question. We trust that any remaining issues can be resolved with the assistance of the journal's editorial team.

Some numbers written in the text do not follow the TC style guide (heps://www.thecryosphere. net/submission.html#math). One example is "6 existing lead products" (L498).
Thanks for pointing this out. We went through the manuscript once more and tried to follow the guidelines. We hope that, in case there are remaining issues, we can sort them out with the help of the editorial team of the journal.

Not all (e.g., …) references in the manuscript include the necessary comma after "e.g." (e.g., L43, L45 and L143)
Done

L142: Citation format is not correct.
Changed

L226: LKLKF is not formatted in the right way.
Done

L336: Citation format is not correct.
Done

L371: There is a space missing between the word "studies" and the start of the citation.
Done

L393-394: There is a comma between the number and the unit in three cases.
Done

L419: L should be replaced with DL.
We have rewritten this section substantially. L is for the typical spatial scale and thus we think it is intuitive to use only "L".

Figure 4 and Figure 5: The caption should probably be "mean (± standard deviation)
Thanks. Yes, we changed it as you suggested.